# Trifluoroacetate induced small-grained CsPbBr$_3$ perovskite films result in efficient and stable light-emitting devices

Haoran Wang[1], Xiaoyu Zhang[2,3], Qianqian Wu[1], Fan Cao[1], Dongwen Yang[2,7], Yuequn Shang[4], Zhijun Ning[4], Wei Zhang [2], Weitao Zheng[2], Yanfa Yan [5], Stephen V. Kershaw[3], Lijun Zhang [2], Andrey L. Rogach [3,6] & Xuyong Yang [1]

Quantum efficiencies of organic-inorganic hybrid lead halide perovskite light-emitting devices (LEDs) have increased significantly, but poor device operational stability still impedes their further development and application. All-inorganic perovskites show better stability than the hybrid counterparts, but the performance of their respective films used in LEDs is limited by the large perovskite grain sizes, which lowers the radiative recombination probability and results in grain boundary related trap states. We realize smooth and pinhole-free, small-grained inorganic perovskite films with improved photoluminescence quantum yield by introducing trifluoroacetate anions to effectively passivate surface defects and control the crystal growth. As a result, efficient green LEDs based on inorganic perovskite films achieve a high current efficiency of 32.0 cd A$^{-1}$ corresponding to an external quantum efficiency of 10.5%. More importantly, our all-inorganic perovskite LEDs demonstrate a record operational lifetime, with a half-lifetime of over 250 h at an initial luminance of 100 cd m$^{-2}$.

[1] Key Laboratory of Advanced Display and System Applications of Ministry of Education, Shanghai University, 149 Yanchang Road, Shanghai 200072, China. [2] Key Laboratory of Automobile Materials of Ministry of Education, State Key Laboratory of Superhard Materials, and College of Materials Science, Jilin University, Changchun 130012, China. [3] Department of Materials Science and Engineering, and Centre for Functional Photonics (CFP), City University of Hong Kong, Kowloon, Hong Kong SAR. [4] School of Physical Science and Technology, ShanghaiTech University, 393 Middle Huaxia Road, Pudong, Shanghai 201210, China. [5] Department of Physics and Astronomy and Wright Center for Photovoltaics Innovation and Commercialization, The University of Toledo, Toledo, OH 43606, USA. [6] Beijing Institute of Technology, School of Materials Science and Engineering, Beijing 100081, China. [7] Present address: Department of Physics and Engineering, Zhengzhou University, Henan 450001, China. These authors contributed equally: Haoran Wang, Xiaoyu Zhang. Correspondence and requests for materials should be addressed to A.L.R. (email: andrey.rogach@cityu.edu.hk) or to X.Y. (email: yangxy@shu.edu.cn)

Metal halide perovskites have recently attracted a lot of attention as promising materials for solution-processed light-emitting devices (LEDs), owing to their excellent optical and electronic properties, such as high charge-carrier mobilities[1,2], saturated emission colors[3,4], and easy color tunability[5,6]. Especially, organic–inorganic 'hybrid' lead bromide perovskites have enabled a major breakthrough in the external quantum efficiency (EQE) for green emitting perovskite LEDs (PeLEDs), which has recently exceeded 20%[7–10]. Unfortunately, hybrid perovskites, which contain small organic cations such as methylammonium (MA) or formamidinium (FA), are extremely moisture-sensitive, which leads to a rapid degradation in LED performance[11–13] and thus limits their prospects for practical applications. Perovskites based on inorganic cesium cations, namely $CsPbX_3$ (X=Cl, Br, and I), exhibit better thermal and chemical stability compared to their hybrid analogues[14,15], and may thus provide the base for high-performance LEDs with reasonable operational stability[16–18]. However, previously reported all-inorganic PeLEDs exhibited relatively poor electroluminescence (EL) performance, suffering mainly from grain boundary related trap states and large crystal size[14,19–21]. Trap states at the adjacent grain boundaries induce severe non-radiative energy transfer, resulting in short photoluminescence (PL) lifetimes and low PL quantum yields (QY) of perovskite films[22], which is detrimental to PeLED performance. Larger perovskite crystal size commonly lowers the probability for radiative recombination, which decreases the EL efficiency of PeLEDs[23,24]. Several strategies have been developed in order to reduce the crystal size in $CsPbBr_3$ films, including anti-solvent vapor treatment, polymer additive assisted film growth, and precursor solution composition optimization, which have enhanced the EQE values of inorganic PeLEDs from less than 1% to over 16% meanwhile[17,25–30]. However, most of those strategies do not allow for a proper control of the grain boundary related trap states, as the reducing of the crystal size also generate more grain boundaries and thus a higher density of trap states in inorganic perovskite films. In addition, the ion migration in polycrystalline perovskite films can occur across the grain boundaries under LED operation[31], which may induce non-radiative recombination in the perovskite emitting layer[32], limiting the PeLED performance and reducing the device stability. Recently, Zeng and co-workers controlled the perovskite nanocrystal surface through employing organic–inorganic hybrid ligands, and have demonstrated $CsPbBr_3$ nanocrystals-based LEDs with an impressive peak EQE of over 16%, revealing the important role of the crystal surface[30].

In this work, we have developed an efficient fabrication approach leading to dense, smooth, and pinhole-free $CsPbBr_3$ perovskite films with high thermal stability, whose grain boundaries are well passivated in order to achieve not only a substantial LED performance enhancement but a greatly improved stability. The inorganic perovskite films are fabricated via one-step solution coating using cesium trifluoroacetate (TFA), as the cesium source instead of the commonly used cesium bromide (CsBr). We have found that the interaction of $TFA^-$ ($CF_3COO^-$) anions with $Pb^{2+}$ cations in the $CsPbBr_3$ precursor solution greatly improves the crystallization rate of perovskite films. By means of theoretical calculations, we show that $TFA^-$ ions are preferentially bound to the surface $Pb^{2+}$ ions in the $CsPbBr_3$. Compared to the conventional CsBr route, the CsTFA-derived films show a flatter energy landscape (a more homogeneous energy level distribution for charges), more stable crystal structure, better optical properties, and suppressed ion migration. As a result, we are able to realize efficient and stable PeLEDs, with a maximum current efficiency of 32.0 cd A$^{-1}$ and a peak EQE of 10.5%, which increases to even higher EQE of 17% for TFA-

derived mixed-cation $FA_{0.11}MA_{0.10}Cs_{0.79}PbBr_3$ LEDs. Even more importantly, the CsTFA-derived PeLEDs show a record half-lifetime of over 250 h at an initial luminance of 100 cd m$^{-2}$, which is almost 17 times longer than that of the CsBr-derived PeLEDs.

## Results

**Preparation and structural analysis of $CsPbBr_3$ films.** Precursor solutions were prepared by mixing different molar ratios, $a$, namely 1.3, 1.5, 1.7, and 1.9, of CsX with $PbBr_2$ in dimethyl sulfoxide (DMSO), where X is either Br$^-$ or TFA$^-$. In the further discussions, the notation 'CsTFA($a$)' for the respective samples will indicate that the CsTFA to $PbBr_2$ molar ratio is $a$, and the notation 'CsBr($a$)' means the CsBr to $PbBr_2$ molar ratio is $a$, respectively. The CsTFA molecule can be considered as consisting of two parts: a Cs$^+$ cation and a TFA$^-$ anion, whose molecular structure is shown in Supplementary Figure 1. The strong electronegativity of TFA$^-$ anions makes it easy for CsTFA to get ionized in polar solvents, which guarantees a high solubility of CsTFA in DMF/DMSO, thus providing abundant Cs$^+$ cations for perovskite formation (Supplementary Note 1). The solubility of CsBr is on the other hand rather limited, making it challenging to obtain high-quality $CsPbBr_3$ films. The radius of TFA$^-$ anion (2.38 Å) is larger than that of Br$^-$ (1.96 Å), making it difficult to dope TFA$^-$ into the $CsPbBr_3$ crystal structure; at the same time, the O=C−O− groups of TFA$^-$ can easily bound to Pb[33], which is helpful for the formation of perovskite films consisting of $CsPbBr_3$ crystals whose surfaces are passivated by TFA$^-$ anions, as we demonstrate in this work. The strong electron pulling ability of F results in the uniform distribution of electrons within the TFA$^-$ anions, which is helpful for the overall stability of this molecular structure, resulting in stable $CsPbBr_3$ films with an enhanced device performance. These are the factors allowing us to realize dense, smooth, and pinhole-free $CsPbBr_3$ perovskite films with high thermal stability, whose grain boundaries are well passivated in order to achieve not only a substantial LED performance enhancement, but a greatly improved stability. $CsPbBr_3$ films were deposited on indium tin oxide (ITO) glass substrates already coated with a layer of poly(3,4-ethylenedioxythiophene): polystyrene sulfonate (PEDOT:PSS) by one-step solution coating of the precursor solutions, followed by thermal annealing under nitrogen atmosphere, as schematically illustrated in Fig. 1a.

Films produced from the CsBr(1.7) precursor without thermal annealing appear colorless and highly transparent (Supplementary Figure 2), indicating that the formation of $CsPbBr_3$ perovskite is inhibited owing to the stability of CsBr•PbBr$_2$•DMSO complexes[33–35]. The strong interaction between DMSO with Pb$^{2+}$ ions and Cs$^+$ ions is evidenced from the Fourier transform infrared (FTIR) spectroscopy (Fig. 1b). For the bare DMSO, the S=O stretching vibration frequency appears at 1040 cm$^{-1}$. This S=O bond vibration shifts to 1014 cm$^{-1}$ for powdered CsBr•PbBr$_2$•DMSO, indicating that the bond strength between sulfur and oxygen decreases due to the formation of CsBr•PbBr$_2$•DMSO complexes[34]. After the thermal treatment at 80 °C, the CsBr-derived films appear yellow in color, as the intermediate phase of CsBr•PbBr$_2$•DMSO complexes converts into $CsPbBr_3$ due to the release of the residual DMSO. In contrast, the films produced from CsTFA(1.7) precursor become yellowish already at room temperature, due to the formation of the $CsPbBr_3$ phase. This observation is consistent with the X-ray diffraction (XRD) patterns of the films measured immediately after deposition (Fig. 1c). The CsBr-derived films exhibit a broad and unstructured XRD profile, meaning that no perovskite crystallization occurred at room temperature; in contrast, CsTFA-

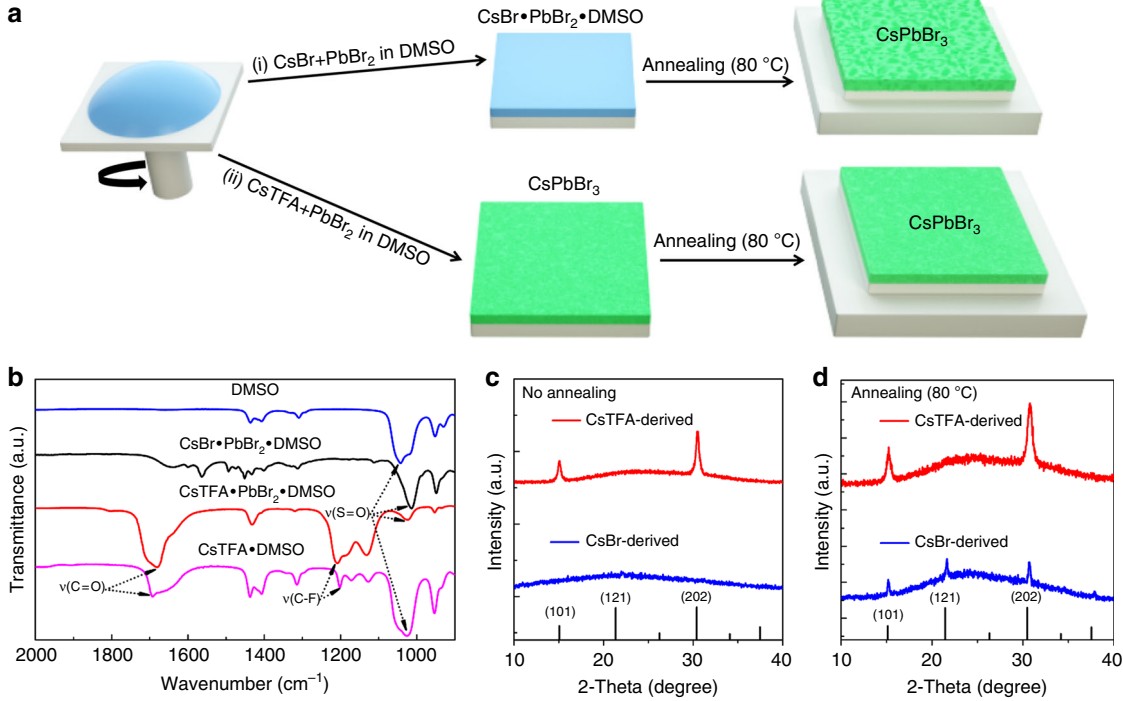

**Fig. 1** FTIR and XRD study of the CsBr- and CsTFA-derived CsPbBr$_3$ perovskites. **a** Schematic presentation of the film fabrication procedure. Route (i) employs commonly used precursors CsBr and PbBr$_2$, while the route (ii) employs CsTFA in place of CsBr. **b** FTIR of DMSO (liquid), CsBr•PbBr$_2$•DMSO (powder), CsTFA•PbBr$_2$•DMSO (powder), and CsTFA•DMSO (powder). **c** XRD patterns of CsPbBr$_3$ films deposited on PEDOT:PSS coated ITO glass from CsBr(1.7) and CsTFA(1.7) without annealing. **d** XRD patterns of CsPbBr$_3$ films deposited on PEDOT:PSS coated ITO glass from CsBr(1.7) and CsTFA(1.7)

derived film shows characteristic diffraction peaks suggesting the formation of a crystalline perovskite phase.

The room temperature crystallization of CsTFA-derived perovskite films can be caused by the strong interaction between TFA$^-$ anions and Pb$^{2+}$ cations in the precursor solution[36]. As shown in Fig. 1b, the FTIR peaks located at 1697 cm$^{-1}$ and 1201 cm$^{-1}$ represent the C=O and CF$_3$ bond stretching vibration of TFA$^-$, respectively[37]. The C=O bond shifts to a lower frequency (1677 cm$^{-1}$) in the complex adduct of CsTFA•PbBr$_2$•DMSO owing to the interaction between the C=O and Pb$^{2+}$ ions[30,38]. The C–F bond shifts to higher frequency (1211 cm$^{-1}$); this may be caused by the higher electron density due to the coordination of the adjacent C=O with Pb$^{2+}$ ions. Note that, the S=O stretching mode wavenumber (1028 cm$^{-1}$) for CsTFA•PbBr$_2$•DMSO is higher than that for CsBr•PbBr$_2$•DMSO (1012 cm$^{-1}$). Such an increase indicates that the interaction between DMSO and Pb$^{2+}$ ions in the former samples is weaker, and thus the formation of perovskite crystals becomes more favorable.

Figure 1d shows XRD patterns of the annealed CsPbBr$_3$ films deposited from CsBr(1.7) and CsTFA(1.7) precursor solutions. Both samples show distinctive XRD peaks at 15.3° and 30.7°, which can be indexed to the (101) and (202) planes for the orthorhombic *Pnma* CsPbBr$_3$ phase[14,16], respectively. For the CsTFA-derived film, the apparent disappearance of the (121) reflection indicates that a preferred orientation effect is probably related to the TFA$^-$ ions favoring the growth of the perovskite crystallites along a certain crystallographic direction. It can also be seen that the crystallinity of the CsTFA-derived film is much higher than that of the CsBr-derived film, which results in a twofold increase in the XRD reflection intensity. Notably, the full-width at half-maximum (FWHM) of the diffraction peaks of CsTFA-derived film is broader, pointing to a decrease of the perovskite crystal grain size. As shown in the corresponding atomic force micrographs (AFMs) (Supplementary Figure 3), the CsTFA-derived film features smaller grain sizes

indeed, and shows a significant reduction in the root mean square (r.m.s) roughness, namely 2.5 nm as compared to 17.5 nm for the CsBr-derived film. All these findings reveal that CsTFA has a beneficial effect on the crystallization of CsPbBr$_3$ from solution.

**Morphologies of CsPbBr$_3$ films**. The morphologies of the spin-cast CsPbBr$_3$ films derived from CsBr(1.7) and CsTFA(1.7) precursor solutions have been investigated by field-emission scanning electron microscopy (FE-SEM). As shown in Fig. 2a, a rather poor surface coverage with large grains is observed for the CsBr-derived film, which is in good agreement with previous reports[14]. In contrast, CsTFA-derived films are uniformly coated on the substrate with an absence of pin-holes (Fig. 2b). From the two high-magnification images shown as insets of Fig. 2, the perovskite grain size for the CsTFA-derived film is greatly decreased from an average value of 300 nm to 60 nm compared with that of the CsBr-derived film, which is consistent with XRD and AFM data. This size reduction indicates that the nucleation density is much higher for the CsTFA-derived films, leading to formation of a large amount of relatively small crystals fully covering the substrate. The cross-sectional view SEM images (Fig. 2c, d) show that both CsBr- and CsTFA-derived films are ~200 nm thick.

To reveal eventual changes in the morphology of the perovskite films as trends, they were deposited from the precursor solutions with varying molar ratios of CsBr:PbBr$_2$ and CsTFA:PbBr$_2$ (Supplementary Figure 4 and Supplementary Note 2). CsTFA-derived films become smoother, and the grain size gradually decreased upon increasing the molar ratio of CsTFA:PbBr$_2$ (Supplementary Figure 4a–d), which further confirms the beneficial role of TFA$^-$ ions in enhancing the density of CsPbBr$_3$ nuclei. In contrast, the increase in CsBr content of the precursor solution does not change the morphology and grain size of the resulting CsPbBr$_3$ films (Supplementary Figure 4e–h).

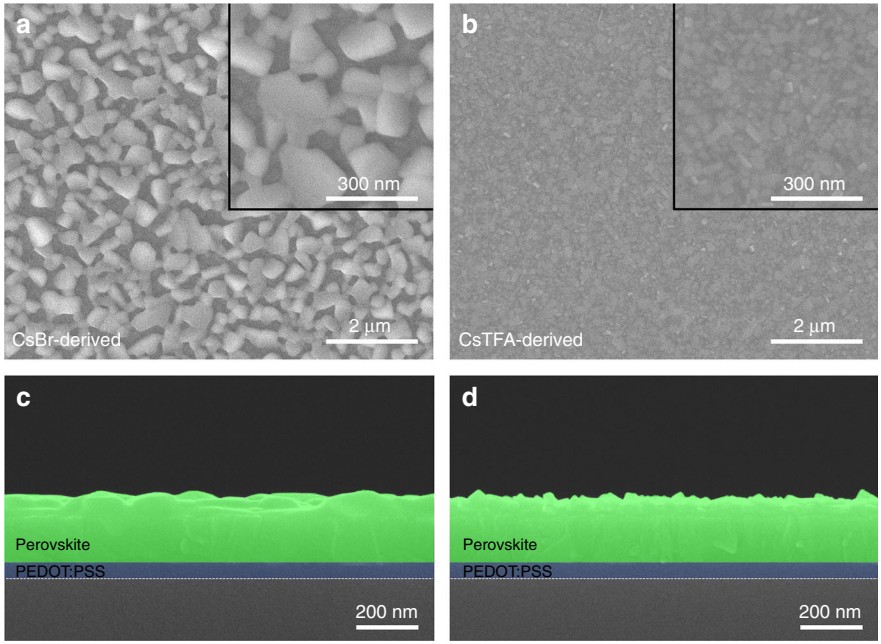

**Fig. 2** Morphologies of the CsBr- and CsTFA-derived CsPbBr$_3$ perovskite films. Top-view **a**, **b** and cross-sectional view **c**, **d** FE-SEM images of CsPbBr$_3$ perovskite films deposited on ITO/PEDOT:PSS substrates from **a**, **c** CsBr(1.7) and **b**, **d** CsTFA(1.7) precursor solutions. Insets in **a** and **b** show high magnification top-view images of the respective films

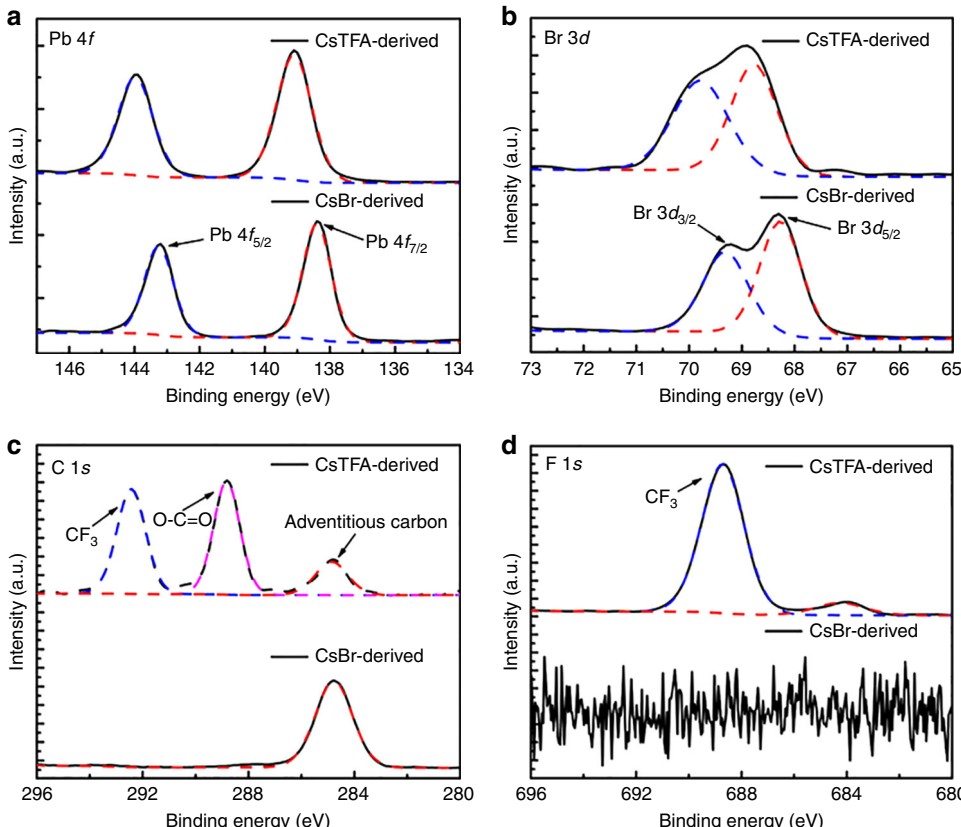

**Fig. 3** XPS characterization. High-resolution XPS spectra of the CsBr(1.7)- and CsTFA(1.7)-derived films for **a** Pb 4f, **b** Br 3d, **c** C 1s and **d** F 1s elements

**Chemical states of CsPbBr$_3$ films.** X-ray photoelectron spectroscopy (XPS) measurements were performed to identify the composition and chemical states of the CsPbBr$_3$ films derived from CsBr(1.7) and CsTFA(1.7). Figure 3a, b shows the XPS spectra for bromine and lead, corrected using the maximum of

the adventitious C 1s signal at 284.8 eV. The Pb 4f spectrum for the CsBr-derived film exhibits two dominant peaks located at 138.4 and 143.2 eV, corresponding to the Pb $4f_{7/2}$ and Pb $4f_{5/2}$ level, respectively. The binding energy peaks of Br 3d at 68.3 eV (Br $3d_{5/2}$) and 69.3 eV (Br $3d_{3/2}$) are also observed in the Br 3d

spectrum for the CsBr-derived film. Notably, a shift of the peak position to higher binding energy is clearly seen for the CsTFA-derived film for Pb 4f and Br 3d. Such shifts in binding energy are attributed to a more stable crystal structure of the formed perovskites, as will be discussed in more detail later. From density functional theory (DFT) calculations (Supplementary Figures 5, 6, Supplementary Tables 1, 2 and Supplementary Note 3), once the TFA⁻ ions are doped into the perovskite crystal structure, the bandgap of the CsPbBr₃ film should exhibit an increase proportionate to the TFA concentration. However, our experimental data show that the bandgap does not widen when the TFA⁻ ions are introduced. Therefore, we conclude that TFA⁻ ions are located at the grain boundaries instead of being doped into the crystal structure of the perovskite. The shifts observed in the XPS spectra originate from the interaction between the TFA⁻ ions and CsPbBr₃ rather than a lattice contraction of perovskite after TFA incorporation, which leads to shorter Pb-Br bonds and higher binding energy for Pb 4f and Br 3d[39,40].

The presence of TFA in the perovskite films has been further confirmed from the C 1s and F 1s XPS spectra. In the C 1s spectrum (Fig. 3c), the peaks at 292.5 and 288.8 eV correspond to bonds in the −CF₃[41], and −O−C=O groups[42], respectively. For the F 1s (Fig. 3d), the peaks at 688.7 eV can be assigned to −CF₃[43]. In addition, energy-dispersive X-ray (EDX) mapping of the CsTFA-derived films shows homogeneous distribution of the F element from TFA (Supplementary Figure 7).

**Optical properties of CsPbBr₃ perovskite films**. Figure 4a, b shows absorption and PL spectra of CsBr(1.7)-derived and CsTFA(1.7)-derived perovskite films, respectively. The absorption edge of the CsTFA-derived film is slightly blueshifted with respect to the CsBr-derived one. Accordingly, a slight blueshift of the PL maximum is also observed in the emission spectra, from 519 nm for the CsBr-derived film to 517 nm for the CsTFA-derived one. Since the crystal grain sizes for both samples (300 nm and 60 nm for the CsBr(1.7)- and CsTFA(1.7)-derived films, respectively) are much larger than the effective Bohr diameters for bulk CsPbBr₃ (7 nm)[44], no quantum confinement effects are expected. At the same time, the CsTFA-derived film has a much higher PL intensity as compared to the CsBr-derived one, with the absolute PLQY increasing from 19% to 57%, which can be seen readily by the naked eye in the photo provided in inset of Fig. 4b.

**Transient absorption spectroscopy analysis**. In addition to the three times improvement in PLQY and the 2 nm blueshift in PL spectra, we also observed the narrowing of the FWHM of the PL peak from 18 nm for the CsBr-derived film to 16 nm for the CsTFA-derived one (Fig. 4b), indicating that the degree of energetic disorder in the latter film may be reduced. To further verify this assumption, and to investigate the effect of the TFA ion treatment on the defect states of the perovskite, the spectral distributions of photoexcited carriers in the perovskite films were studied by ultrafast transient absorption (TA) spectroscopy. After the perovskite films absorb photons, the photoexcited carriers relax to the lowest energy sites, which appears as a redshift of the transient bleach minimum[45]. A 1.6 meV redshift of the transient bleach was observed for the CsTFA-derived film; whereas, the shift for CsBr-derived film was ~5.6 meV (Fig. 4c–e, and Supplementary Figure 8). The reduced redshift indicates that a flatter energy landscape and decreased band-tail states are observed in the CsTFA-derived film, suggesting its promising future for LEDs, with a minimized charge transport barrier and a maximized radiative transition probability.

Since the TFA⁻ ions are shown to be located at the grain boundaries of the perovskite films, the reduced band-tail states

should be mainly due to the improved passivation of the crystal surface defects. Time-resolved PL spectroscopy (Fig. 4f) shows that the average carrier lifetime ($\tau_{avg}$) reaches 71 ns for the CsTFA (1.7)-derived film, which is a 1.5-fold increase as compared to that for the CsBr(1.7)-derived film (46 ns) (Supplementary Note 4). Both the PLQY and $\tau_{avg}$ progressively increase upon raising the molar ratio of CsTFA:PbBr₂ (Supplementary Figure 9 and Supplementary Table 3). The blueshifted emission peak of the CsTFA-derived film also indicates the efficient passivation of defects, since spontaneous radiative recombination taking place over trap states often leads to a longer emission wavelength compared to that from the band edge transition[46]. From the combination of data on the PL peak positions, PLQYs and the PL lifetimes, we conclude that the TFA⁻ ions bound to the perovskite crystal surface have passivated grain boundary related trap states successfully.

**Grain boundary passivation mechanism**. The schematic diagram of grain boundary passivation mechanism was given in Fig. 5. Although some surface defects have been passivated by TFA⁻, there are still some trap states left in the grain boundaries inferred by the non-single exponential PL decay curves. The grain boundaries with trap states for pure CsPbBr₃ films have a side effect on the PL emission. The trap states can attract both electrons and holes and thus cause non-radiative recombination at grain boundaries, as shown by the band diagram in Fig. 5a. In contrast, the TFA⁻ passivated grain boundary regions formed a larger bandgap than the particle interior (PI), which pushes both electrons and holes away from the grain boundaries and drift into PIs (Fig. 5b). We characterized the surface potential of the TFA-derived perovskite film by Kelvin Probe Force Microscopy (KPFM), and found that the test results consist well with our suggested energy scheme. The perovskite film surface is directly accessible by the probe to measure the contact potential difference. Supplementary Figure 10a shows the topography of the film surface, and Supplementary Figure 10b shows the contact potential difference. Some individual grains are clearly distinguishable, and grain boundaries have higher surface potential than that of the particle interior, which pushes charge carriers away from the grain boundaries and drifts them into particle interior. The topography, contact potential, and their overlapped 3D maps of a $3 \times 3\,\mu m^2$ area (Supplementary Figure 11a, b, and c) and a $1 \times 1\,\mu m^2$ area (Supplementary Figure 11d, e, and f) provide information of the contact potential difference between the grain boundaries and the crystals. We note that our contact potential data are different from those previously reported of the pure perovskites, whose grain boundaries typically have lower contact potential values than that within the grains[47]. This is because the TFA⁻ anions are abundant at the grain boundaries and can push the charges into the grains. This process leads to a significant enhancement in radiative recombination of electrons and holes. In addition, the calculated exciton binding energy ($E_B$) values are 65.5 meV for the CsTFA-derived film and 50.7 meV for the CsBr-derived film (Supplementary Figure 12). The higher $E_B$ for the CsTFA-derived film is mainly due to the smaller crystal sizes, and the formation of larger bandgap/smaller bandgap (grain boundary/grain) structures. The charge carriers within the CsTFA-derived films are easier to bound together and form excitons, which greatly enhances the PL QY and the device EL efficiency[24,48].

**Thermal stability of CsPbBr₃ perovskite films**. The thermal stability of perovskite films is an important factor for their possible application in LEDs. We compared the emission spectra of CsBr(1.7)- and CsTFA(1.7)-derived films in the temperature

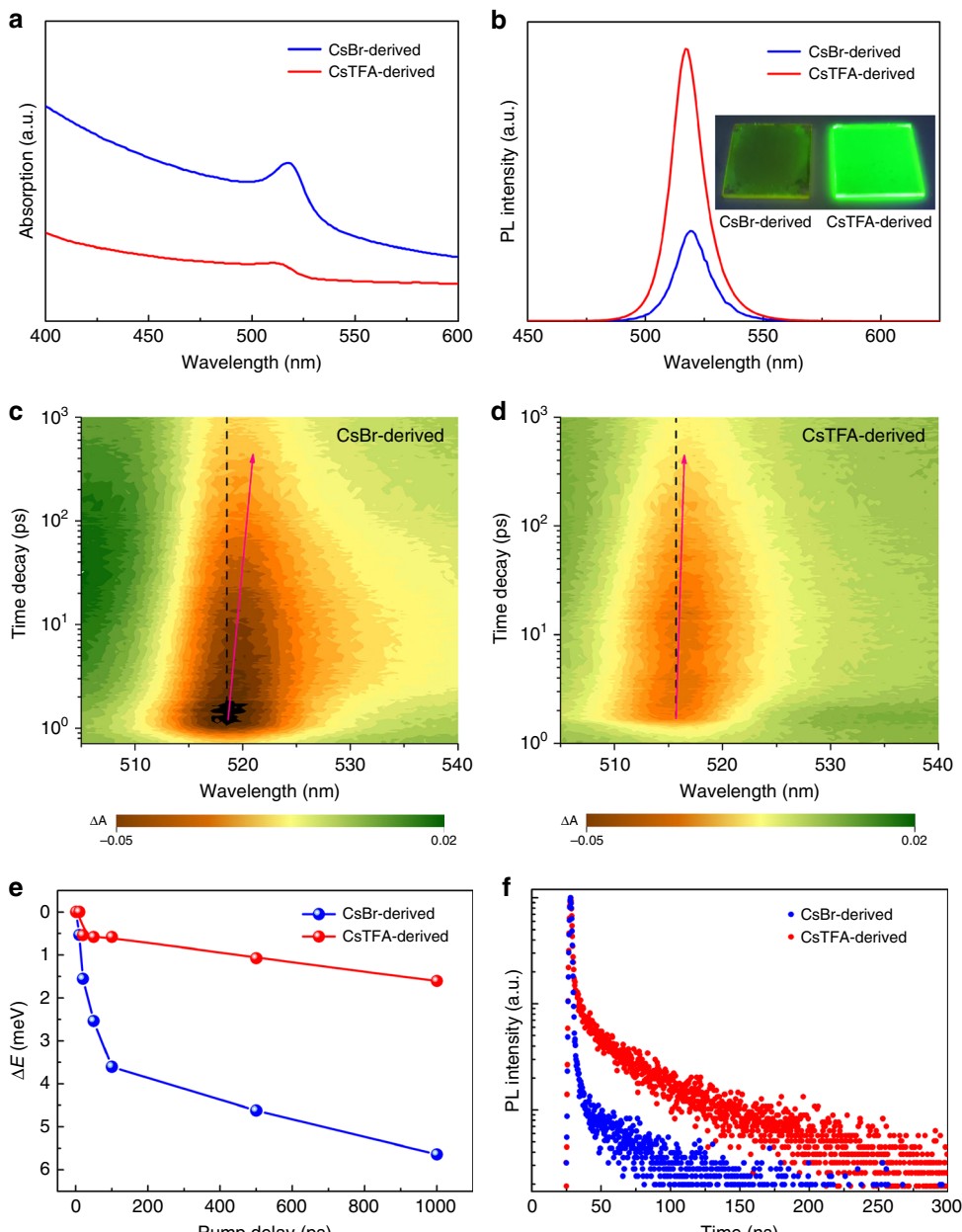

**Fig. 4** Photophysical properties of the CsBr- and CsTFA-derived CsPbBr$_3$ perovskite films. **a** Absorption and **b** PL spectra of the CsBr(1.7)- and CsTFA(1.7)-derived CsPbBr$_3$ perovskite films. Inset in **b** shows the photograph of the green-emitting perovskite films, taken under UV lamp excitation. **c**, **d** Transient absorption maps of the CsBr(1.7)-derived and CsTFA(1.7)-derived films, respectively. Pump wavelength and pump pulse width are 365 nm and 100 fs, respectively. The red arrows indicate the trajectory of the bleach minimum position. ΔA represents optical density. **e** Shifts of the bleach minimum position over time for the two types of the CsBr(1.7)- and CsTFA(1.7)-derived films. **f** Time-resolved PL decay curves of the CsBr(1.7)- and CsTFA(1.7)-derived films on ITO/PEDOT:PSS substrates

range from 80 to 200 °C. As shown in Supplementary Figure 13, the relative PL intensity of the CsBr(1.7)-derived film is strongly decreased to 5% of the initial value when the temperature is increased to 200 °C. In contrast, the PL intensity of the CsTFA (1.7)-derived film kept 54% of its original emission intensity after annealing at 200 °C, which reveals it has a much better thermal stability than that of the CsBr(1.7)-derived film. We attribute this improvement to a more stable crystal structure for the CsTFA (1.7)-derived film.

**Device structure and performance.** PeLEDs based on the CsTFA-derived perovskite films were fabricated. Figure 6a shows

the schematic device architecture of a typical PeLED, with a multi-layer structure consisting of indium tin oxide (ITO)/poly (3,4-ethylenedioxythiophene):polystyrene sulfonate (PEDOT: PSS)/CsPbBr$_3$/1,3,5-tris(1-phenyl-1H-benzimidazol-2-yl)benzene (TPBI)/LiF/Al. PEDOT:PSS is used as the hole injection/transport layer (HIL/HTL), and TPBI serves as the electron transport layer (ETL). The ITO and LiF/Al act as anode and cathode, respectively. The corresponding schematics of the flat-band energy-level diagram of the device is shown in Fig. 6b, with the energy levels for the CsTFA-derived CsPbBr$_3$ films calculated from ultraviolet photoelectron spectroscopy (UPS) and optical data (Supplementary Figures 6, 14 and Supplementary Table 4). The EL spectrum of the PeLEDs based on the CsTFA(1.7)-derived CsPbBr$_3$ films

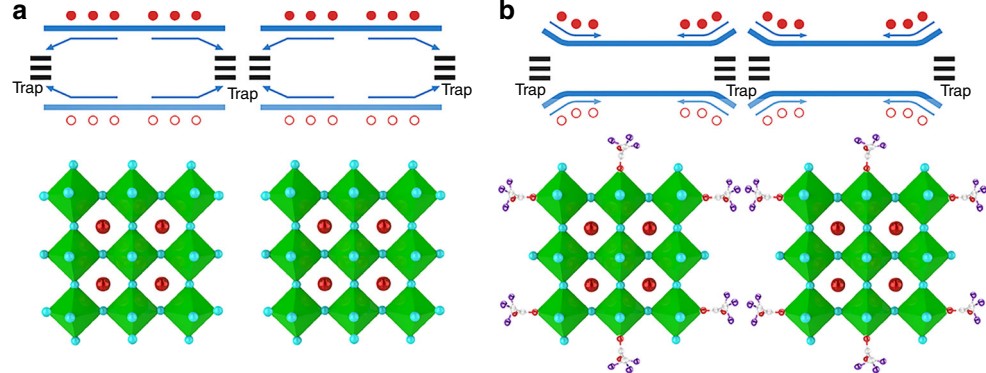

**Fig. 5** Grain boundary passivation mechanism. Schematic diagram of grain boundary passivation mechanism for **a** CsBr- and **b** CsTFA-derived films

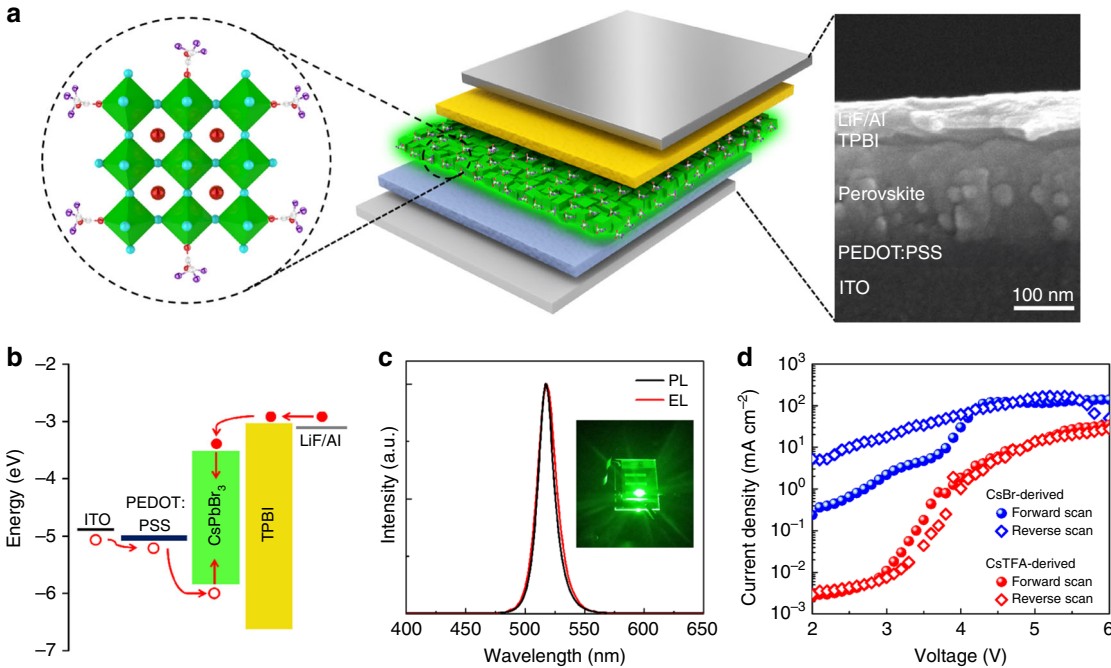

**Fig. 6** Device structure of PeLEDs. **a** Device structure and a corresponding cross-sectional TEM image of the multi-layer PeLEDs: ITO/PEDOT:PSS (40 nm)/CsPbBr₃ (200 nm)/TPBI (20 nm)/LiF (1 nm)/Al (100 nm). **b** Schematic flat-band energy diagram of the PeLED; the energy levels for PEDOT:PSS and TPBI are taken from reference[6]. **c** Normalized PL spectrum of the CsPbBr₃ film, and EL spectrum of the PeLED at an applied voltage of 5.5 V. Inset: The photograph of the operating device with an emitting area of 2 mm × 2 mm. **d** J–V hysteresis of PeLEDs based on CsBr(1.7)- and CsTFA(1.7)-derived CsPbBr₃ perovskite films

exhibits an emission peak at 518 nm, with a narrow FWHM of 19 nm (Fig. 6c). Compared to the PL emission wavelength of the respective perovskite film, the EL peak is slightly broadened and redshifted by ~1 nm, which most probably results from the electric-field-induced Stark effect[49,50]. The photograph given in the inset of Fig. 6c displays a bright green emission from an operating PeLED at an applied voltage of 5.5 V.

The large photocurrent hysteresis in perovskite solar cells and PeLEDs induced by the trap states on the surface and grain boundaries in perovskite films is one of the major hindrances impairing both the performance and the stability of the respective devices[31,51]. To investigate the photocurrent hysteresis in our PeLEDs, their current density–voltage (J–V) curves were measured in various scanning directions (Fig. 6d). It is found that PeLEDs based on the CsBr(1.7)-derived perovskite films indeed show a hysteresis between the forward and the reverse scans. In contrast, the devices based on the CsTFA(1.7)-derived film show a relatively low hysteresis, indicating that TFA⁻ ions

bonded to the perovskite crystal surface have successfully passivated the grain boundary related trap states, and the ion migration through grain boundaries has been suppressed. The PL evolution over time under a constant voltage (2 V, below the turn-on voltage) and the EL evolution over applied bias for the TFA-derived CsPb(Br/I)₃ LEDs further prove this assumption (Supplementary Figures 15, 16).

PeLEDs based on CsBr- and CsTFA-derived perovskite films with varying molar ratios of precursors were optimized as presented in Supplementary Figure 17 and Supplementary Tables 5,6; it was found that the devices with the best performance were those based on CsBr(1.7)- and CsTFA(1.7)-derived films, respectively. Figure 7a–d provides the comparison of the J–V, luminance–voltage (L–V), current efficiency–voltage (CE–V), and external quantum efficiency–voltage (EQE–V) characteristics of the PeLEDs based on CsBr(1.7)- and CsTFA(1.7)-derived films. From the J–V curves, a higher current density was observed for the CsBr-derived film, which is due to its lower surface coverage (Fig. 2a) leading to

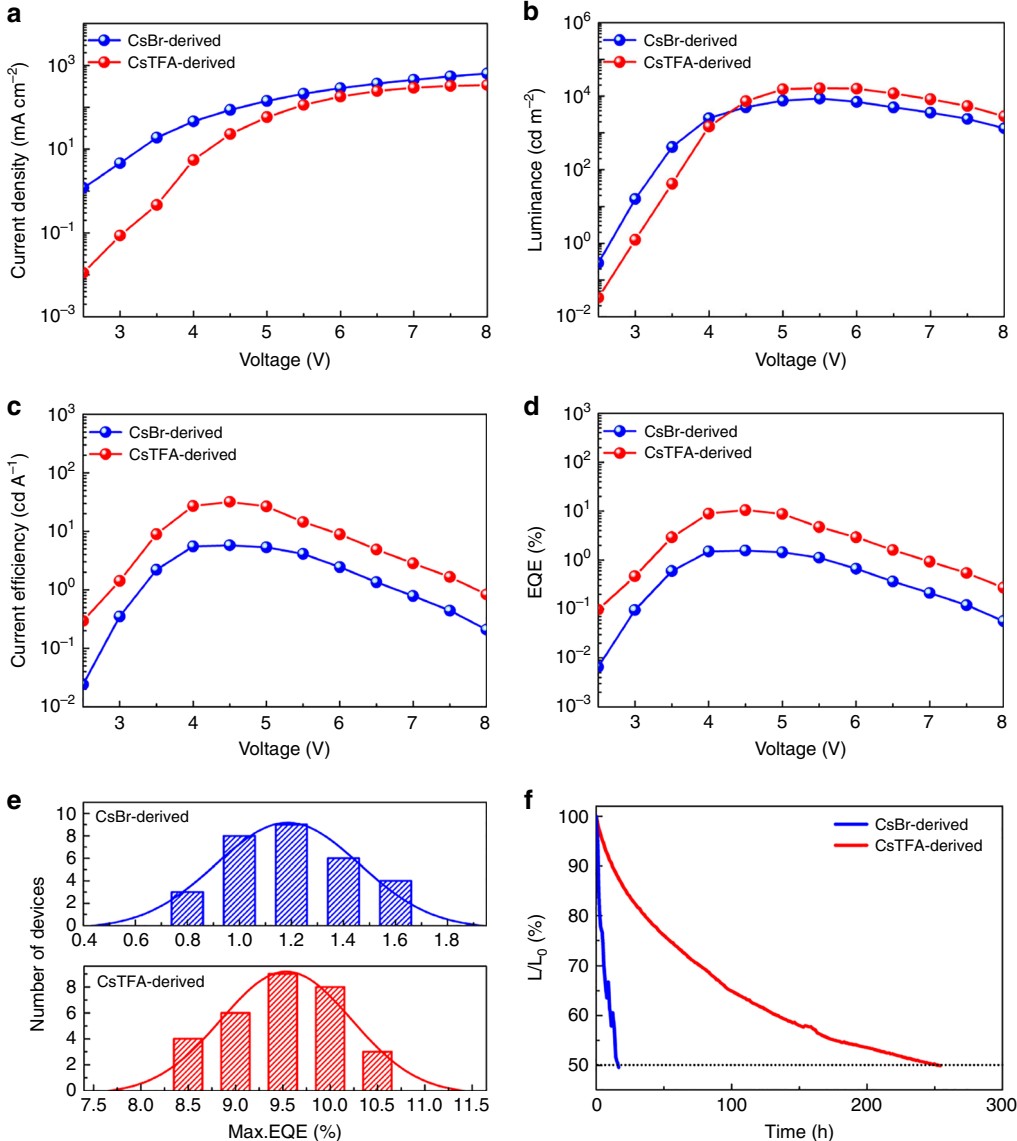

**Fig. 7** Device performance of the CsB- and CsTFA-derived PeLEDs. **a** *J–V*, **b** *L–V*, **c** CE–*V*, **d** EQE–*V* curves, **e** performance reproducibility, and **f** operational lifetimes of the PeLEDs based on CsBr(1.7)- and CsTFA(1.7)-derived CsPbBr$_3$ perovskite films

electrical shunting paths. The turn-on voltages for the two types of devices were close, namely 2.8 V for the CsTFA-derived device and 2.5 V for the CsBr-derived one. However, the maximum luminance (16,436 cd m$^{-2}$) of the PeLED based on the CsTFA-derived perovskite film was two times higher than that of the CsBr-derived one (8628 cd m$^{-2}$). The maximum CE (32.0 cd A$^{-1}$) and EQE (10.5%) of the PeLEDs based on the CsTFA-derived films were much higher than those of the CsBr-derived ones (maximum CE 5.78 cd A$^{-1}$, and maximum EQE 1.6%). Figure 7e provides the histogram of the maximum EQE values for 30 devices based on CsBr- and CsTFA-derived films, operated under respective optimal conditions. The average maximum EQEs for the devices based on CsBr- and CsTFA-derived films were 1.2% and 9.4%, respectively. We note, that compared to the CsBr-derived devices, the CsTFA-derived ones show significantly lower standard deviation (5.4% versus 19.9%), demonstrating the good performance reproducibility of these PeLEDs. Besides, TFA-derived mixed-cation FA$_{0.11}$MA$_{0.10}$Cs$_{0.79}$PbBr$_3$ LEDs were fabricated and have shown a maximum luminance of 35,700 cd m$^{-2}$, and a high peak EQE of 17%, further proving the beneficial effect of CsTFA (Supplementary Figure 18).

Operational stability is of crucial importance for PeLEDs. The device lifetime comparison for the PeLEDs based on CsBr- and CsTFA-derived perovskite films was tested in air, using simple ultraviolet-curable epoxy resin encapsulation (Fig. 7f). PeLEDs based on the CsBr-derived films show a rapid deterioration from an initial luminance of 100 cd m$^{-2}$, with a half-lifetime of only 15 h. In contrast, CsTFA-based PeLEDs show a greatly improved half-lifetime of up to 250 h, exceeding a number of recent reports[7–9,48,52,53]. Such a significant improvement in the device stability is attributed to the realization of smooth, pinhole-free CsPbBr$_3$ perovskite films with TFA passivated grain boundaries and enhanced thermal stability, which reduces Joule heating caused by poor surface coverage of the inorganic perovskite emissive layer and impedes ion motion through the grain boundaries.

## Discussion

To conclude, we have shown that TFA$^-$ ions are able to passivate perovskite grains, which enables formation of smooth, pinhole-free CsPbBr$_3$ perovskite films with high PLQY of 57%, leads to more stable crystal structure, and flatter energy landscapes. TFA$^-$

anions interact with $Pb^{2+}$ ions in the precursor solution, inducing the fast crystallization of small-grained $CsPbBr_3$ perovskite crystals during the one-step film deposition. The $TFA^-$ ions were found to be located at the grain boundaries, and bound to the perovskite crystal surface, which efficiently passivates the surface defects, and decreases the perovskite crystal grain size (from 300 nm to 60 nm) resulting in high radiative transition probability (radiative rates of the best CsTFA-derived devices were 1.9 times higher than the corresponding best CsBr-derived devices). As a result, favorable EL performance for PeLEDs has been achieved, with a maximum CE of 32.0 cd $A^{-1}$, a peak EQE of 10.5%, and a 17 times improvement in operational lifetime compared with CsBr-derived PeLED. TFA-derived mixed-cation $FA_{0.11}MA_{0.10}Cs_{0.79}PbBr_3$ LEDs showed even higher peak EQE of 17%, further proving the beneficial effect of CsTFA on the different types of both all-inorganic and hybrid perovskite films. Our study suggests that the high color-purity and low-cost all-inorganic lead halide perovskite films can be developed into highly efficient and stable LEDs via carefully optimizing the grain boundaries.

## Methods

**Materials**. PEDOT:PSS (Clevious PVP AI 4083) was purchased from Heraeus. TPBI and LiF were purchased from Luminescence Technology. Cesium trifluoroacetate and $PbBr_2$ were purchased from Alfa and Sigma-Aldrich, respectively. All materials were used as received.

**Perovskite precursor solutions**. Cs$X$ ($X$ is Br, or trifluoroacetate (TFA)) and $PbBr_2$ were dissolved in anhydrous DMSO at a molar ratio ranging from 1.3 to 1.9, with final solids concentrations of ~12 wt%. Mixtures were stirred at 60 °C overnight and filtered through polytetrafluoroethylene filters (0.45 μm pore size) before using.

**PeLED fabrication**. Patterned ITO substrates were sequentially sonicated with detergent, deionized water, acetone, and isopropyl alcohol for 15 min in each solvent. After drying under nitrogen flow, the substrates were treated for 15 min with an oxygen plasma. A PEDOT:PSS layer was spin-coated onto the ITO substrates at a speed of 4000 rpm for 40 s and then baked at 150 °C for 10 min in air. To form a perovskite layer, the perovskite precursor solution was spin-coated onto the PEDOT:PSS layers in a glove box, and baked at 80 °C for 10 min. The samples were transferred into a vacuum chamber, and TPBI (20 nm), LiF (1 nm), and Al (100 nm) layers were sequentially deposited by thermal evaporation under a pressure of less than $4 \times 10^{-4}$ Pa.

**Characterization**. FT-IR spectra were recorded under attenuated total reflectance mode on a Spectrum 100 (Perkin Elmer) spectrometer. XRD patterns were obtained on a Bruker D8 Advance diffractometer with Cu Kα radiation over an angular range from 10° to 60° at a scanning rate of 6° min$^{-1}$. AFM, KPFM, and C-AFM measurements were performed using a Bruker Dimension Icon microscope. FE-SEM measurements were conducted on a JEOL JSM-7500F microscope. Absorption spectra were measured on a Perkin Elmer Lambda 950 UV–vis–NIR spectrometer. PL steady state, time-resolved spectra and PL QY data were collected on an Edinburgh FLS920 PL spectrometer. The temperature-dependent PL spectra were measured through a Newton CCD (model no. DU920P-BU) integrated with Shamrock spectrometer (model no. SR-750-D1-R) with a laser excitation (442 nm) at different temperatures in a helium cryostat (CRYO Cool-G2B-LT). UPS and XPS were analyzed on a Thermo Scientific Escalab 250Xi. UPS measurements utilized the He (I) photo line (21.22 eV) from a He discharge lamp and the high-binding energy secondary electron cutoff ($E_{cutoff}$), and the HOMO region data were extracted from the UPS spectra. The HOMO levels could be expressed as HOMO = 21.22 − ($E_{cutoff}$ − $E_\Delta$) (where $E_\Delta$ is the gap between the HOMO level and the Fermi level ($E_F$)). Transient absorption measurements were conducted on an ExciPro femtosecond transient absorption pump-probe spectrometer (CDP Systems Corp). Femtosecond laser pulses were generated by an Astrella ultrafast Ti: Sapphire amplifier at a 1 kHz repetition rate. Pump wavelength and pump pulse width are 365 nm and 100 fs, respectively. The current density–luminance–voltage ($J$–$L$–$V$) characteristics were measured on a Keithley 2400 source meter. EL spectra of PeLEDs were measured on a PR-670 Spectra Scan spectroradiometer. The device lifetime tests of PeLEDs were performed on a ZJZCL-1 OLED ageing lifespan test instrument.

**First principles calculations**. Calculations were performed within the framework of density functional theory (DFT) by using plane-wave pseudopotential methods as implemented in the Vienna Ab initio Simulation Package[54,55]. The electron–ion interactions were described by the projected augmented wave pseudopotentials[56]

with the 1$s$ (H), 2$s^2$ and 2$p^2$ (C), 2$s^2$ and 2$p^3$ (N), 2$s^2$ and 2$p^4$ (O), 2$s^2$ and 2$p^5$ (F), 5$s^2$5$p^6$6$s$ (Cs), 4$s^2$ and 4$p^5$ (Br) and 5$d^{10}$, 6$s^2$ and 6$p^2$ (Pb) electrons-treated explicitly as valence electrons. We used the generalized gradient approximation formulated by Perdew, Burke, and Ernzerhof[57] as the exchange correlation functional. Kinetic Energy cutoff for the plane-wave basis set were set to 400 eV. The k-point meshes with grid spacing of $2\pi \times 0.03$ Å$^{-1}$ were used for electronic Brillouin zone integration. The structures of $CsPbBr_3$ with different contents of TFA doping were optimized through total energy minimization with the residual forces on the atoms converged to below 0.01 eV Å$^{-1}$. To properly take into account the long-range van der Waals interactions that play a nonignorable role in the hybrid perovskites involving organic molecules, the vdW-optB86b functional[58] was adopted.

## Data availability

The data that support the findings of this study are available from the corresponding author upon reasonable request.

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

## Acknowledgements

The authors acknowledge the financial support from the National Natural Science Foundation of China (51675322, 61605109, 61735004, and 51702115), National Key Research and Development Program of China (2016YFB0401702), National Natural Science Foundation of China/Research Grants Council Joint Research Scheme (N_CityU108/17), Shanghai Rising-Star Program (17QA1401600), The Program for Professors of Special Appointment (Eastern Scholar) at Shanghai Institutions of Higher Learning (TP2015037), National Postdoctoral Program for Innovative Talents (BX201600060), the NSFC/RGC project N_CityU108/17, and the Talent Introduction Plan of Overseas Top Ranking Professors by the State Administration of Foreign Expert Affairs (MSBJLG040).

## Author contributions

X.Y. and A.L.R. proposed the research direction and guided the project. H.W., Q.W. and F.C. fabricated and characterized the PeLED devices. D.W., L.Z. and S.V.K. carried out density functional theory calculations. H.W., X.Z., D.Y., Y.S., Z.N., L.Z., W.Z (Zhang)., W.Z (Zheng)., Y.Y., S.V.K., A.L.R. and X.Y. analyzed and discussed the experimental results. H.W. and X.Z. provided equal contributions to this study. All authors contributed to the manuscript.

## Additional information

**Competing interests:** The authors declare no competing interests.

