## [Peer Review File · Nature Communications]

Editorial Note: Parts of this peer review file have been redacted as indicated to remove third-party material where no permission to publish could be obtained.

Reviewers' comments:

Reviewer #1 (Remarks to the Author):

In the manuscript, the authors have provided new kind of anion molecules for efficient and ultra-stable all-inorganic perovskite light-emitting diodes (PeLEDs). The authors showed astonishingly long operation lifetime of PeLEDs with significantly enhanced EL efficiency. Also, the paper is well written and organized, so we recommend the publication of this paper after minor revision with a few comments as follows.

1. The device performance and analysis of chemical and structural properties with CsTFA were well provided. But still, the reason for choosing the specific molecule is unclear. Can the authors strengthen the explanation about molecular structure and chemical property of it?
2. According to Fig 5, the TFA- passivated grain boundary regions should have the larger bandgap than the bulk part, making similar energy diagram with quantum well structure. Still, the authors should provide much solid evidence to present such an energy scheme. We recommend the authors to strengthen the story by Ultraviolet Photoelectron Spectroscopy (UPS) or surface potential measurement (Contact potential difference, conductive AFM ...).
3. Can the authors provide the temperature-dependent photoluminescence (PL) result to compare the exciton binding energy of the perovskite films with or without CsTFA?
4. The authors showed greatly enhanced EL lifetime of CsTFA-derived PeLEDs, but the origin of lifetime enhancement is still unknown. Can the authors provide any possible degradation mechanisms which were prevented with CsTFA and the experimental evidence for it? For example, if the authors can conduct temperature-dependent conductivity measurement, Arrhenius fitting on the conductivity will show evidence on the activation energy for ion migration.
5. The format for reference citation is not consistent. Please make sure the reference list set in the same format.

Reviewer #2 (Remarks to the Author):

In this paper, the author reports efficient and stable inorganic perovskite-based light-emitting devices with max. EQE of 10.5% and lifetime of 250 hours at an initial luminance of 100 cd/m² by introducing the trifluoroacetate anions to passivate the surface effects and control the crystal size. Unfortunately, I do not think this MS merits publication in Nature Communications.

There are many reports of high efficiency perovskite-based LEDs now. For example, the EQE of over 15% reached by additive Nature Communications 9, 3892 (2018); the CsPbBr₃ LEDs with EQE > 15% is published in ACS Nano 12, 9541 (2018); the CsPbBr₃ QLEDs with EQE of 11.6 is published in Adv. Mater. 30, 1100764 (2018); the EQE of 16.3% for perovskite LEDs is published in ACS Nano 12, 8808 (2018); even, the CsPbBr₃ film-based LEDs with EQE of 10.4% is realized by controlling the composition published in Nature Communications 8, 15640 (2017). Thus, the EQE demonstrated here is not so efficient compared with these previous reports, even low. Meanwhile, these reports exhibited the trifluoroacetate anions were not efficient to enhance the device performance.

The lifetime is 250 hours in the MS. They think "the lifetime is almost 17 times longer than that of the CsBr-derived PeLEDs". But they may omitted the ref. (Adv. Funct. Mater. 2017, 1700338). In the ref. the device of all-inorganic CsPbBr₃ PeLED exhibits only 12% degradation after 80 h of continuous operation at an initial luminance of 1000 cd/m², meaning the lifetime (t₈₈) is 80 hours at 1000 initial luminance of 1000 cd/m². Here, in the MS the lifetime is operated at initial luminance of 100 cd/m², lifetime (t₅₀) is 250 hours. Thus, the lifetime in this MS is not obviously improved.

In addition, they have found that the TFA-anions can greatly improve the crystallization rate of perovskite films. The formation of perovskite crystal became more favorable after adding the FTA, which demonstrate the reaction become fiercer. The description is conflict with the small crystal

formation. TFA-anions do not effectively inhibit the ion migration, or the reaction of formation of the perovskite crystal is so easy. If the TFA-anions can decrease the grain size, the PL and absorbance should exhibit blue shift after the increase of the content TFA. But the PL in Fig S8a exhibit an opposite tendency. Thus, the TFA can decrease the grain size and inhibit the ion migration is not so convincing.

In the MS, the description of "which have enhanced the EQE values of PeLEDs from less than 1% to 5%" is so obsolete and not proper, now the EQE is over 16%, even 20%.

Reviewer #3 (Remarks to the Author):

In this manuscript, the authors used Cs salt (CsTFA) not the traditional CsBr as the Cs source for synthesizing CsPbBr₃ film and fabricating light-emitting diodes (LEDs). The results showed that the new Cs salt has delivered higher device efficiency (10.5% EQE), and better operational stability (>250 hours). Generally, I think the efficiency shown in this manuscript is not attractive, while the stability achieved is very meaningful because the stability of PeLED is the main concern of this area and the published results just showed several hours stability. The paper could be further considered after addressing the following issues.

1) For traditional CsPbBr₃ formation, it is the mixture of CsBr and PbBr₂, the formation equation could be $\text{CsBr} + \text{PbBr}_2 \rightarrow \text{CsPbBr}_3$, no matter how the CsBr excess shown in this manuscript. While for the CsTFA proposed in this study, $\text{CsTFA} + \text{PbBr}_2 \rightarrow \text{CsPbBr}_3$??? The CsTFA cannot provide any Br source, how the chemical reaction could lead to CsPbBr₃ formation?

2) The authors claimed the TFA- existed in grain boundary based on the density functional theory (DFT) calculations. Is there any experimental proof to confirm it? And the authors argued that the device using CsTFA precursor could suppress the ion migration in the perovskite layer, the authors should provide some experimental results.

3) As the authors shown in Figure 7f, the devices using CsTFA showed much better operational stability. While from the Figure S11 b and d, these two devices showed almost the same drop off, even the CsTFA based devices showed more serious drop off. As we known, the drop off could reflect the stability of the devices. On this condition, why the CsTFA showed better operational stability?

Reply to Reviewer #1

In the manuscript, the authors have provided new kind of anion molecules for efficient and ultra-stable all-inorganic perovskite light-emitting diodes (PeLEDs). The authors showed astonishingly long operation lifetime of PeLEDs with significantly enhanced EL efficiency. Also, the paper is well written and organized, so we recommend the publication of this paper after minor revision with a few comments as follows.

Reply:

Many thanks for the reviewer's positive comments.

Comment 1#. *The device performance and analysis of chemical and structural properties with CsTFA were well provided. But still, the reason for choosing the specific molecule is unclear. Can the authors strengthen the explanation about molecular structure and chemical property of it?*

Reply:

Our rationale for choosing this particular molecule can be summarized as follows. The CsTFA molecule (CF_3COOCs) can be considered as consisting of two parts: the Cs^+ cations and the TFA^- anions (CF_3COO^-). The molecular structure of TFA^- is given in **Figure R1**. The strong electronegativity of TFA^- anions makes it easy for CsTFA to get ionized in polar solvents, which guarantees a high solubility of CsTFA in DMF/DMSO, thus providing abundant Cs^+ cations for perovskite formation. The

solubility of CsBr is on the other hand rather limited, making it challenging to obtain high-quality CsPbBr₃ films. The size of TFA⁻ anion (2.38 Å) is larger than that of Br⁻ (1.96 Å), making it difficult to dope TFA⁻ into the CsPbBr₃ crystal structure; at the same time, the O=C-O- groups of TFA⁻ can easily bound to Pb (*Adv. Mater.* **2018**, **30**, **1800764**), which is helpful for the formation of perovskite films consisting of CsPbBr₃ crystals whose surfaces are passivated by TFA⁻ anions, as we demonstrate in this work. The strong electron pulling ability of F results in the uniform distribution of electrons within the TFA⁻ anions, which is helpful for the overall stability of this molecular structure, resulting in stable CsPbBr₃ films with an enhanced device performance. These are the factors allowing us to realize dense, smooth, and pinhole-free CsPbBr₃ perovskite films with high thermal stability, whose grain boundaries are well passivated in order to achieve not only a substantial LED performance enhancement but a greatly improved stability. Related discussion has been added to the revised manuscript on **Pages 5 and 6**.

Figure R1. Molecular structure of TFA⁻ anion. Elements color coding: O (red), C (grey), and F (purple).

Q2: According to Fig 5, the passivated grain boundary regions should have the larger bandgap than the bulk part, making similar energy diagram with quantum well

structure. Still, the authors should provide much solid evidence to present such an energy scheme. We recommend the authors to strengthen the story by Ultraviolet Photoelectron Spectroscopy (UPS) or surface potential measurement (Contact potential difference, conductive AFM ...).

Reply:

We characterized the surface potential of the TFA-derived perovskite film by kelvin probe force microscopy (KPFM), and found that the test results consist well with our suggested energy scheme. The perovskite film surface is directly accessible by the AFM probe to measure the contact potential difference. **Figure R2a** shows the topography of the film surface, and **Figure R2b** shows the contact potential difference. Some individual grains are clearly distinguishable, and grain boundaries have higher surface potential than that of the particle interior, which pushes charge carriers away from the grain boundaries and drifts them into particle interior. The topography, contact potential, and their overlapped 3D maps of a $3\times 3\ \mu\text{m}^2$ area (**Figure R3a, b, and c**) and a $1\times 1\ \mu\text{m}^2$ area (**Figure R3d, e, and f**) provide information of the contact potential difference between the grain boundaries and the crystals. We note that our contact potential data are different from those previously reported of the pure perovskites, whose grain boundaries typically have lower contact potential values than that within the grains (*J. Phys. Chem. Lett.* 2015, 6, 875–880). This is because the TFA⁻ anions are abundant at the grain boundaries and can push the

charges into the grains. Related discussion has been added to the revised manuscript on Pages 16, 41, and 42.

Figure R2. (a) Topography map and (b) contact potential difference image of the TFA-derived CsPbBr₃ films.

Figure R3. (a,d) Topography, (b,e) contact potential difference, and (c,f), their overlap 3D images for differently sized TFA-derived CsPbBr₃ films.

Q3: Can the authors provide the temperature-dependent photoluminescence (PL) result to compare the exciton binding energy of the perovskite films with or without CsTFA?

Reply:

The CsBr(1.7) and the CsTFA(1.7) samples were chosen for the temperature-dependent photoluminescence (PL) measurement to compare the exciton binding energy difference, as shown in **Figure R4**. We determined the exciton binding energy (E_B) using the following equation:

$$I(T) = \frac{I_0}{1 + Ae^{(-E_B/k_B T)}}$$

where I_0 is the emission intensity at 0 K, A is a scaling factor, and k_B is the Boltzmann constant. The calculated E_B values are 65.5 meV for the CsTFA-derived film and 50.7 meV for the CsBr-derived film. The higher E_B for the CsTFA-derived film is mainly due to the smaller crystal sizes, and the formation of larger bandgap/smaller bandgap (grain boundary/grain) structures. The carriers within the CsTFA-derived films are easier to bound together and form excitons, which greatly enhances the PL QY and the device EL efficiency. Related discussion has been added to the revised manuscript on **Pages 16, 17, 42, 43**.

Figure R4. Integrated PL emission intensity as a function of temperature from 80 to 300 K for CsTFA-derived and CsBr-derived films.

Q4: The authors showed greatly enhanced EL lifetime of CsTFA-derived PeLEDs, but the origin of lifetime enhancement is still unknown. Can the authors provide any possible degradation mechanisms which were prevented with CsTFA and the experimental evidence for it? For example, if the authors can conduct temperature-dependent conductivity measurement, Arrhenius fitting on the conductivity will show evidence on the activation energy for ion migration.

Reply:

Just as the reviewer mentioned, the introduction of CsTFA to perovskite films suppresses the ion migration, which plays an important role in improving the device stability performance. There was already a strong evidence in our manuscript on this point, which is the reduced hysteresis of the TFA-derived CsPbBr₃ LEDs (**Figure R5**). We do not have access to the equipment which would allow us to perform

temperature-dependent conductivity measurements, but following the reviewer's suggestion, we have developed another set of experiments to further proof this particular point of the suppressed ion migration, such as described in the following. Since the bandgap changes over the halogen ratios for mixed-halide perovskites, we can determine the components based on the emission color. Thus, to provide a direct observation of ion migration, the mixed-halide CsPb(Br/I)₃ perovskites were taken, and their emission was traced in the LED structure of ITO/PEDOT:PSS/CsPb(Br/I)₃/TPBI/LiF/Al. First, the PL evolution over time under a constant voltage (2V, below the turn-on voltage) was investigated. As shown in **Figure R6**, the PL spectra experience 15 and 6 nm red-shifts occurring within 3 min for CsBr- and CsTFA-derived CsPb(Br/I)₃ devices, respectively. The stronger PL shift of the CsBr-derived device indicates more intense ion migration within the film. In addition, the PL completely disappeared for the CsBr-derived device in 9 min, most probably due to destruction of perovskite structure or the formation of defects related to the halogen deficiency. In contrast to that, the CsTFA-derived CsPb(Br/I)₃ device was still shining brightly, because the ion migration was suppressed here. Besides, when the applied bias increased from 5 to 8 V, the EL peaks experienced 12 and 4 nm red-shifts for CsBr- and TFA-derived CsPb(Br/I)₃ perovskite LEDs, respectively (**Figure R7**), which further demonstrates that the ion migration has been suppressed with the help of TFA ions. Related discussion has been added to the revised manuscript on **Pages 19, 44, and 45**.

Figure R5 (Fig. 6d of our revised manuscript). *J-V* hysteresis of PeLEDs based on CsBr(1.7)- and CsTFA(1.7)-derived CsPbBr₃ perovskite films.

Figure R6. Normalized PL spectra of the CsBr- and CsTFA-derived CsPb(Br/I)₃ devices under a constant bias of 2 V. PL spectra were recorded from 0 min to 3 min.

Figure R7. Normalized EL spectra of (a) the CsBr- and (b) CsTFA-derived CsPb(Br/I)₃ devices. EL spectra were recorded from 5V to 8V.

Q5: *The format for reference citation is not consistent. Please make sure the reference list set in the same format.*

Reply:

All reference citations have been corrected to become fully consistent.

Reply to Reviewer #2

In this paper, the author reports efficient and stable inorganic perovskite-based light-emitting devices with max. EQE of 10.5% and lifetime of 250 hours at an initial luminance of 100 cd/m² by introducing the trifluoroacetate anions to passivate the surface effects and control the crystal size. Unfortunately, I do not think this MS merits publication in Nature Communications. There are many reports of high efficiency perovskite-based LEDs now. For example, the EQE of over 15% reached by additive Nature Communications 9, 3892 (2018); the CsPbBr₃ LEDs with EQE > 15% is published in ACS Nano 12, 9541 (2018); the CsPbBr₃ QLEDs with EQE of 11.6 is published in Adv. Mater. 30, 1100764 (2018); the EQE of 16.3% for perovskite LEDs is published in ACS Nano 12, 8808 (2018); even, the CsPbBr₃ film-based LEDs with EQE of 10.4% is realized by controlling the composition published in Nature Communications 8, 15640 (2017). Thus, the EQE demonstrated

here is not so efficient compared with these previous reports, even low. Meanwhile, these reports exhibited the trifluoroacetate anions were not efficient to enhance the device performance. The lifetime is 250 hours in the MS. They think “the lifetime is almost 17 times longer than that of the CsBr-derived PeLEDs”. But they may omitted the ref. (Adv. Funct. Mater. 2017, 1700338). In the ref. the device of all-inorganic CsPbBr₃ PeLED exhibits only 12% degradation after 80 h of continuous operation at an initial luminance of 1000 cd/m², meaning the lifetime (t₈₈) is 80 hours at 1000 initial luminance of 1000 cd/m². Here, in the MS the lifetime is operated at initial luminance of 100 cd/m², lifetime (t₅₀) is 250 hours. Thus, the lifetime in this MS is not obviously improved. In addition, they have found that the TFA⁻anions can greatly improve the crystallization rate of perovskite films. The formation of perovskite crystal became more favorable after adding the TFA, which demonstrate the reaction become fiercer. The description is conflict with the small crystal formation. TFA-anions do not effectively inhibit the ion migration, or the reaction of formation of the perovskite crystal is so easy. If the TFA-anions can decrease the grain size, the PL and absorbance should exhibit blue shift after the increase of the content TFA. But the PL in Fig S8a exhibit an opposite tendency. Thus, the TFA can decrease the grain size and inhibit the ion migration is not so convincing. In the MS, the description of “which have enhanced the EQE values of PeLEDs from less than 1% to 5%” is so obsolete and not proper, now the EQE is over 16%, even 20%.

Reply:

We respectfully disagree with this referee's analysis, and we provide our grounds in the reply below, based on the following point-to-point comparison with the mentioned published work. The referee raised several points in his/her comments, including: 1) whether the device EQE is high enough; 2) whether the device stability is sufficiently enhanced than previously reported results, 3) whether the TFA-derived CsPbBr₃ films should have larger crystal sizes, and 4) whether the ion migration in TFA-derived CsPbBr₃ films is indeed suppressed. Six papers have been mentioned to support some of these points, which we are listing below as follows:

Ref.1: Nature Communications 9, 3892 (2018)

Ref.2: ACS Nano 12, 9541 (2018)

Ref.3: Adv. Mater. 30, 1100764 (2018), which should be Adv. Mater. 30, 1800764 (2018)

Ref.4: ACS Nano 12, 8808 (2018)

Ref.5: Nature Communications 8, 15640 (2017);

Ref.6: Adv. Funct. Mater. 2017, 1700338, which should be Adv. Funct. Mater. 2017, 27, 1700338

In addition, two recently published related papers are also listed as **Ref.7** and **Ref. 8** here:

Ref. 7: Nature 562, 245, (2018)

Ref. 8: Adv. Mater. DOI: 10.1002/adma.201805409

We have made a summary of relevant published characteristics of these perovskite LEDs, including **our own device**, which is given in **Table R1**.

Perovskite type	Peak EQE (η_p , %)	Average EQE (η_a , %)	Operation Condition and Lifetime	Ref.
PEA ₂ Cs _{n-1} Pb _n Br _{3n+1} film	15.5	12.5	At 2 mA/cm ² , L ₅₀ ≈ 90 min	1
PPABr-CsPbBr ₃ quantum dots	15.17	14	At 2.5 mA/cm ² , L ₅₀ ≈ 72 min	2
FA _x Cs _{1-x} PbBr ₃ quantum dots	11.6	11.2	--	3
FAPbBr ₃ nanocrystals	16.3	7.3	At 4 V, L ₅₀ < 30 sec	4
Cs _{0.87} MA _{0.13} PbBr ₃ film	10.4	7.3	At 3.7 V, L ₅₀ < 40 sec	5
PEO-CsPbBr ₃ film	4.76	~3.9	~1000 cd/m ² , L ₈₀ ≈ 80 h	6
CsPbBr ₃ /MABr quasi-core/shell structure	20.3	--	100 cd/m ² , L ₅₀ = 104.56 h	7
CsPbBr ₃ quantum dots	16.48	15.1	At 0.6 mA/cm ² , L ₅₀ = 136 min	8
CsPbBr ₃ film	10.5	9.4	100 cd/m ² , L ₅₀ > 250 h	Our work

We first address the first comment of the reviewer: whether our device's EQE is high enough. We do agree with the referee that the LED performance is an important parameter to show that a new method or a device structure is efficient. However, such a comparison is little grounded when it is made to compare apples with oranges. Our all-inorganic CsPbBr₃ emitters are different from most of the materials in the listed papers (**Ref. 1, 2, 3, 4, and 5**), since four of them (**Ref. 1, 2, 3, and 5**) have been A-site doped with larger organic cations, and the other one (**Ref. 4**) is organic-inorganic hybrid FAPbBr₃ nanocrystals. Doping with larger A-site cations

will make the tolerance factor closer to 1, resulting in more stable device structure and less lattice distortion related trap states (*Scientific Reports. 2016, 6, 23592*). Introducing organic component into inorganic perovskites will change their conductivity from n-type to near-ambipolar, resulting in more balanced charge transport and flatter band conditions under operation. Thus, it is easier to achieve higher performing LEDs with those emitters, and it is not fair to compare them with all-inorganic perovskite LEDs. At the same time, we have recently realized that through doping with larger organic A-site cations, the TFA-derived $\text{FA}_{0.11}\text{MA}_{0.10}\text{Cs}_{0.79}\text{PbBr}_3$ LEDs reach the EQE of 17%, which is higher than all the devices mentioned in the papers listed by the referee (**Figure R8**). This is an encouraging development and continuation of the work reported in our original manuscript, which also nicely emphasizes the overall generality of our reported TFA-related approach.

Figure R8. (a) J - V - L , and (b) CE - J - EQE curves of the CsTFA-derived $\text{FA}_{0.11}\text{MA}_{0.10}\text{Cs}_{0.79}\text{PbBr}_3$ PeLED.

Now let us turn to the second comment, namely about the device stability. The devices from **Ref.6** have shown great operational stability, which was commented by the referee as “our device lifetime is not obviously improved”. However, the device peak EQE in **Ref.6** is 4.76%, much lower than ours. From the perspective of the potential commercialization, our TFA-derived method is more attractive, as it can simultaneously achieve both high device performance and good stability. At this point, we would also like to discuss the stability issue from the other researchers’ point view. In their **Table R2**, taken from recently published paper in Nature (**Ref. 7**), in the stability performance discussion section the authors only listed those perovskite LEDs with peak EQEs over 10%, mentioning specifically that both the device stability and performance are important for LEDs. The LEDs reported in **Ref. 7** have shown the longest T_{50} , which is 104.56h at 100 cd/m²; while our TFA-derived devices achieved even better (nearly 2.5 times) stability performance. We recall at this point that two other reviewers of our original manuscript indicated that our TFA-derived LEDs have shown excellent stability performance, namely “*The authors showed astonishingly long operation lifetime of PeLEDs*” and “*the stability achieved is very meaningful because the stability of PeLED is the main concern of this area*”.

Table R2. The summary from the recently published Nature paper (**Ref. 7**).

[Redacted]

The third comment was about whether the TFA-derived CsPbBr₃ films should have larger crystal sizes. From the referee's point of view, the formation of perovskite crystals became more favorable after adding the TFA, which demonstrate the reaction become fiercer, and thus should form bigger crystals. On the contrary, as previously reported, one can obtain large perovskite crystals 1) via solvent annealing to decrease the crystal growth rate (*Adv. Mater.* **2014**, *26*, **6503**), and 2) via doping to increase the crystal formation energy and thus to decrease the crystal growth rate (*ACS Appl. Mater. Interfaces.* **2017**, *9*, **2403**). This is because when decreasing the perovskite crystal growth rate, the nucleation process results in formation of fewer seed crystals, which would consume more precursors, resulting in larger sized crystals. In comparison, if one increases the crystal growth rate, the nucleation process becomes

fiercer and forms more seed crystals, which would then consume less precursors, resulting in smaller sized crystals. This has also been reported while introducing an antisolvent to increase the crystal growth rate during the perovskite formation, which has also decreased the crystal size (*Science* 2015, 350, 1222). Based on these observations, the conclusion is that TFA can indeed lead to fiercer reaction, but the smaller crystals can also be simultaneously obtained in this case which is beneficial to the increased amount of seed crystals.

1. The fourth comment was that introducing TFA ions cannot suppress the ion migration. We have already provided a reply on this very same question 4 of the reviewer 1, which appeared above. Besides, the referee pointed out that the PL peak doesn't keep blue-shifting when increasing the content of the TFA anions. This is because that even with TFA ions, our perovskite crystal sizes are still much larger than the Bohr diameters of the respective bulk perovskites (*Chem. Mater.* 2017, 29, 3644), and thus no quantum confinement effects are expected. Lastly, the description “*which have enhanced the EQE values of PeLEDs from less than 1% to 5%*” has been revised to outline the true improvement observed.

All the related discussions have been added to the revised manuscript on **Pages 2, 4, 5, 19, 21, 37, 44, 45, and 46.**

Reply to Reviewer #3

In this manuscript, the authors used Cs salt (CsTFA) not the traditional CsBr as the Cs source for synthesizing CsPbBr₃ film and fabricating light-emitting diodes (LEDs). The results showed that the new Cs salt has delivered higher device efficiency (10.5% EQE), and better operational stability (>250 hours). Generally, I think the efficiency shown in this manuscript is not attractive, while the stability achieved is very meaningful because the stability of PeLED is the main concern of this area and the published results just showed several hours stability. The paper could be further considered after addressing the following issues.

Reply:

We are thankful to the reviewer for his overall positive and encouraging comments.

Q1: *For traditional CsPbBr₃ formation, it is the mixture of CsBr and PbBr₂, the formation equation could be CsBr+PbBr₂CsPbBr₃, no matter how the CsBr excess shown in this manuscript. While for the CsTFA proposed in this study, CsTFA+PbBr₂CsPbBr₃???. The CsTFA cannot provide any Br source, how the chemical reaction could lead to CsPbBr₃ formation?*

Reply:

For the preparation of TFA-derived perovskite films, the PbBr₂ and CsTFA were

firstly dissolved in DMSO, ionizing into Pb^{2+} , Br^- , Cs^+ , and TFA^- ions. Due to the ionic nature of the lead-halide perovskites (*Adv. Funct. Mater.* 2016, 26, 2435), the precursors gradually precipitate and grow into crystals as the DMSO evaporated. The formation equation should be $\text{Cs}^+ + \text{Pb}^{2+} + 3\text{Br}^- \rightarrow \text{CsPbBr}_3$ within the grains, and $\text{Cs}^+ + \text{Pb}^{2+} + (3-x)\text{Br}^- + x\text{TFA}^- \rightarrow \text{CsPb}(\text{Br}/\text{TFA})_3$ ($0 \leq x \leq 3$) at the grain boundaries. The related discussion has been added to the revised manuscript on **Pages 35,36**.

Q2: The authors claimed the TFA- existed in grain boundary based on the density functional theory (DFT) calculations. Is there any experimental proof to confirm it? And the authors argued that the device using CsTFA precursor could suppress the ion migration in the perovskite layer, the authors should provide some experimental results.

Reply:

Since the O=C-O- groups can easily bond to the Pb^{2+} ions of perovskites (*Adv. Mater.* 2018, 30, 1800764) as evidenced from the FTIR and XPS results shown in **Figure R9** and **Figure R10**, and the optical bandgap doesn't become narrower for the TFA-derived films (doping larger sized TFA^- into the crystal lattice would narrow the bandgap), we conclude that the TFA^- ions sit at grain boundaries. For the TFA derived films, the TFA^- passivated grain boundary regions form a larger bandgap than that of the particle interior, which pushes both electrons and holes away from the grain boundaries and let them drift into the grains. Thus, the contact potential difference

between the grain boundaries and the grains can help to determine the exact location of TFA⁻ ions. The surface potential of the TFA-derived perovskite film was characterized by kelvin probe force microscopy (KPFM). The film surface is directly accessible by the AFM probe to measure the contact potential difference. **Figure R2a** shows the topography of the film surface, and **Figure R2b** shows the contact potential difference. Some individual grains are clearly distinguishable and grain boundaries have higher surface potential than that of the particle interior, demonstrating that the TFA⁻ does exist in grain boundaries. All related discussions have been added to the manuscript on **Pages 16, 41 and 42**.

Figure R9 (Fig. 1c of our revised manuscript). FTIR of DMSO (liquid), CsBr•PbBr₂•DMSO (powder), CsTFA•PbBr₂•DMSO (powder), and CsTFA•DMSO (powder).

Figure R10 (Fig. 3 of our revised manuscript). High-resolution XPS spectra of the CsBr(1.7)-derived and CsTFA(1.7)-derived films for (a) Pb 4f, (b) Br 3d, (c) C 1s and (d) F 1s elements.

About the ion migration suppression, we addressed the same question of the Reviewer 1 in our reply to his/her question #4, see above. The related discussion has been added to the revised manuscript on **Pages 19, 44, and 45**.

Q3: *As the authors shown in Figure 7f, the devices using CsTFA showed much better operational stability. While from the Figure S11 b and d, these two devices showed almost the same drop off, even the CsTFA based devices showed more serious drop off. As we known, the drop off could reflect the stability of the devices. On this condition, why the CsTFA showed better operational stability?*

Reply:

For the perovskite LEDs, it is true that more pronounced efficiency drop-off means more unbalanced charge injection, resulting in both charge accumulation and the efficiency decrease. However, in our case, we need to analyze the efficiency drop-off and the stability performance from two aspects, namely the emitting layer quality and the charge injection barrier. On one hand, as we see from the UPS data (**Table R3**), the TFA-derived film has a deeper VBM (-5.95 eV) than that of CsBr (-5.87 eV), indicating that the hole injection barrier becomes higher for the TFA-derived LEDs, and thus induces more pronounced efficiency drop-off. On the other hand, as show in **Figure R11a**, the pin-holes in the CsBr-derived films lead to electrical shunting paths and lower the radiative recombination of the emissive layer, which greatly decreases the device operational stability. In comparison, the TFA-derived films are smooth, compact, and pin-hole free, which greatly enhances their stability (**Figure R11b**). That's why even so the TFA-derived LEDs show more obvious efficiency drop-off, they still have better stability performance. It also means that through the proper charge injection optimization or interface engineering, we can further improve the EQE and stability of TFA-derived LEDs.

Table R3 (Table S1 of our revised manuscript). Energy levels of CsBr(1.7)- and CsTFA(1.7)-derived perovskite films.

Samples	VBM (eV)	CBM (eV)
CsBr(1.7)	-5.87	-3.58

CsTFA(1.7)

-5.95

-3.64

Figure R11 (Fig. 2 in our revised manuscript). Top-view images of CsPbBr₃ perovskite films deposited on ITO/PEDOT:PSS substrates from (a) CsBr(1.7) and (b) CsTFA(1.7) precursor solutions. Insets in (a) and (b) show high magnification top-view images of the respective films.

Reviewers' comments:

Reviewer #1 (Remarks to the Author):

In the manuscript, the concerns of the reviewers were adequately addressed in respect of molecular origin of the new precursor's benefit on device performance and luminescent property, and definite evidence on the energy level alignment at the crystal surface after the revision. Also, additional experiment using mixed-halide system clearly showed the suppressed halide ion's migration with CsTFA. We think the manuscript is largely strengthened after the revision and can be accepted by Nature Communications now.

With best regards,

Tae-Woo Lee

Reviewer #2 (Remarks to the Author):

The manuscript reports that the trifluoroacetate can significantly improve the performance of all inorganic perovskite based LEDs (a max. EQE of 10.5%, a half-lifetime of over 250 h at an initial luminance of 100 cd m⁻²). This work presented is interesting and important to some extent, but I think it does not reach the criterion of Nature Communication based on the comments as follows. It lacks significant detail and needs major additions.

As for the previous comments,

a. The author think the result is high in the all inorganic based CsPbBr₃, but the current paper (EQE of >20% , Nature 562, 245, (2018)) shows that the result in the Yang's draft is not so high . In the recent paper (Nature 562, 245, (2018)), Wei et al think the emitter is all inorganic based CsPbBr₃, the MABr is only on the surface of the emitters. Meanwhile, there are plenty of organic TFA in this work, similar to the paper in Nature with organic MABr. However, the above published result is 2-fold higher than this paper.

b. As for the stability, the authors think the result is compatible the EQE with the stability. But the author still misses the result (EQE of 12.98% and the operating lifetimes T50 with an initial brightness of 1000 cd/m² is 173 hours. Thus, T50 @100cd/m² is exceeding 1000 hours in an operating conditions) in the paper (Materials Today Chemistry 10 (2018) 104-111).

In addition, some new comments as follows,

1. I do not think that TFA- ions are located at the grain boundaries instead of being doped into the crystal structure of the perovskite. According to the authors' speculation, EDX mapping of the TFA-perovskite film should accumulate more proportion of F elements at the boundaries of the films, which should not be homogeneous distribution as shown in Figure S7. And the crystal and lattice structure of the film need be further probed. This is particularly important when considering the characterization methods, such as the TEM.

2. Although the author gives some explanation about the stability of the device enhanced by the TFA, the expression is not so convinced. As we all known, the TFA is a short organic materials. Its own stability is not very good, compared to Cs. So the evidence of the enhancement of the stability should be strengthened in the MS.

3. The authenticity of the cross-sectional view is heavily masked by the added color, the contact surface between PEDOT: PSS and perovskite are so smooth. These would misunderstand the readers. The author should revise it.

4. The introduction of FA and MA in perovskite films increased the performance of corresponding devices, while if it can reduce the stability as both MA and FA are extremely moisture-sensitive. Further, please analysis the reason for the development of device performance with FA and MA, especially the beneficial effect of CsTFA.

5. Author says that the charge carriers within the CsTFA-derived films are easier to bound together

and form excitons, but authors do not prove that the excitons are effective for radiative recombination, as excitons may have many behaviors, such as Auger recombination.

6. Whether such a thick perovskite emitter ~200 nm will adversely affect the electrical transport properties of the device.

7. The authors claimed that the PL and EL shifts of mixed-halide CsPb(Br/I)₃ device caused by different time and voltage are attributed to ion migration, and how the authors determined that this phenomenon was not caused by phase separation. Other characterization technique such as TOF need be further probed.

8. This manuscript needs to be revised in more detail, there are still some errors. For example:

i: In the sentence "The strong interaction between DMSO with Pb²⁺ ions and Cs⁺ ions is evidenced from the Fourier transform infrared (FTIR) spectroscopy (Fig. 1b)" (page 6, paragraph 2), Fig.1b should be corrected to Fig1c.

ii: In page 21, "Besides, TFA-derived mixed cation FA_{0.11}MA_{0.10}Cs_{0.79}PbBr₃ LEDs were fabricated and have shown have shown a maximum luminance of 35,700 cd m⁻²", it have double "have shown".

Reviewer #3 (Remarks to the Author):

The authors have almost addressed my concerns, but the explanation for improving the device stability is still not very convincing, especially the response for the Q3 mentioned by the Reviewer 3#. The paper could be accepted for publication in present form, but it could be much better if further improvement could be done.

Reply to Reviewer #1

In the manuscript, the concerns of the reviewers were adequately addressed in respect of molecular origin of the new precursor's benefit on device performance and luminescent property, and definite evidence on the energy level alignment at the crystal surface after the revision. Also, additional experiment using mixed-halide system clearly showed the suppressed halide ion's migration with CsTFA. We think the manuscript is largely strengthened after the revision and can be accepted by Nature Communications now.

With best regards,

Tae-Woo Lee

Reply:

We highly appreciate Prof. Lee's positive comments; it was a pleasure to have you among our reviewers.

Reply to Reviewer #2

The manuscript reports that the trifluoroacetate can significantly improve the performance of all inorganic perovskite based LEDs (a max. EQE of 10.5%, a half-lifetime of over 250 h at an initial luminance of 100 cd m⁻²). This work presented is interesting and important to some extent, but I think it does not reach the criterion of Nature Communication based on the comments as follows. It lacks significant detail and needs major additions.

Reply:

We appreciate these additional comments of the reviewer which have helped us to further improve this manuscript. In the following, we are addressing all these comments and we hope that the reviewer would appreciate our efforts.

a. The author think the result is high in the all inorganic based CsPbBr₃, but the current paper (EQE of >20%, Nature 562, 245, (2018)) shows that the result in the Yang's draft is not so high. In the recent paper (Nature 562, 245, (2018)), Wei et al think the emitter is all inorganic based CsPbBr₃, the MABr is only on the surface of the emitters. Meanwhile, there are plenty of organic TFA in this work, similar to the paper in Nature with organic MABr. However, the above published result is 2-fold higher than this paper.

Reply:

The emitter in Wei's devices was indeed inorganic CsPbBr₃, but of a rather different kind, namely a CsPbBr₃/MABr quasi-core/shell structure (please see the description in their paper, **Figure R1**). The MABr is an ionic compound, which is not very stable. For our case, although the organic TFA is used to improve the quality of inorganic perovskite films and suppress the ion migration, the small molecule TFA with a covalent bond is more stable than the ionic compound MABr. Note that organic LEDs (OLEDs) with small-molecule functional materials have been already commercialized (*Nature communications* **9**, 3207 (2018).). Therefore, our devices show more than 2-fold better stability compared with Wei's devices (250h vs 105h),

and even longer device stability could be obtained by further improving the device performance. Moreover, the efficiency of Wei's devices is not twofold higher than in our work, as we also obtained very similar efficiency by adjusting the perovskite composition (20.3%-EQE for Wei's devices, 10.5%-EQE for our TFA-derived CsPbBr₃ devices and 17.0%-EQE for our TFA-derived FA_{0.11}MA_{0.10}Cs_{0.79}PbBr₃ devices).

[Redacted]

Figure R1 from the Wei's Nature paper.

b. As for the stability, the authors think the result is compatible the EQE with the stability. But the author still misses the result (EQE of 12.98% and the operating lifetimes T_{50} with an initial brightness of 1000 cd/m² is 173 hours. Thus, T_{50} @100cd/m² is exceeding 1000 hours in an operating conditions) in the paper (Materials Today Chemistry 10 (2018) 104-111).

Reply:

Our work is quite different from the above paper (*Materials Today Chemistry 10 (2018) 104-111*). We have focused on the improvement in the device stability by enhancing the film quality of inorganic perovskite emitter in a standard organic-inorganic hybrid device structure. On the contrary, the operating lifetime improvement in the mentioned Teridi's study is by using an all-inorganic device structure (**Figure R2**). Inorganic charge transport materials are generally more stable than organic materials, so that it is unfair to make such a comparison. In addition, in their work, they have used an organic-inorganic hybrid perovskite as emitter and achieved a 12.98%-EQE, while we have obtained a higher EQE of 17% when the organic-inorganic hybrid perovskite ($\text{FA}_{0.11}\text{MA}_{0.10}\text{Cs}_{0.79}\text{PbBr}_3$) is used. Our method so far still possesses distinct merits as compared with Teridi's study.

Figure R2. (a) Teridi's device structure. (b) Our device structure.

In addition, some new comments as follows,

1. I do not think that TFA^- ions are located at the grain boundaries instead of being doped into the crystal structure of the perovskite. According to the authors' speculation, EDX mapping of the TFA-perovskite film should accumulate more proportion of F elements at the boundaries of the films, which should not be

homogeneous distribution as shown in Figure S7. And the crystal and lattice structure of the film need be further probed. This is particularly important when considering the characterization methods, such as the TEM.

Reply:

Due to the resolution limit of SEM, the distribution for the F element may appear homogeneous on the EDX maps of the TFA-derived perovskite films. It is very well known fact that interpretation of the EDX elemental analysis must be taken with a great care, as this method is far from exact. In addition to this single method, we have carefully confirmed the TFA distribution theoretically and experimentally via the density functional theory (DFT) calculation, the Tauc plots, the XRD results, and the kelvin probe force microscopy (KPFM), with a solid conclusion that the TFA⁻ ions are indeed accumulated at the grain boundaries, as we summarize once again in details below.

From the DFT calculations (**Figure R3**), the band gap of the CsPbBr₃ films should increase with the increase of TFA content if the TFA⁻ ions are doped into the perovskite crystal structure. However, our experimental data show that the band gap of perovskite films with TFA is almost the same with that of perovskite films without TFA (**Figure R4**). Moreover, XRD patterns show that no diffraction peak from TFA-derived films shifts to smaller angles, indicating that TFA⁻ ions are also not doped into the perovskite crystal structure ($R_{\text{TFA}} > R_{\text{Br}}$) (**Figure R5**).

Figure R3 from **Fig. S5** of our revised manuscript. Calculated band gaps of $\text{CsPbBr}_{3-x}(\text{TFA})_x$ at different TFA concentrations.

Figure R4 from **Fig. S6** of our revised manuscript. Tauc plots showing the dependence of $(\alpha h\nu)^2$ of perovskite films upon the incident photon energy ($h\nu$) (assuming direct allowed transitions).

Figure R5 from **Fig. 1d** of our revised manuscript. XRD patterns of CsPbBr₃ films deposited on PEDOT:PSS coated ITO glass from CsBr(1.7) and CsTFA(1.7).

The distribution of TFA has been further confirmed by measuring the contact potential difference using kelvin probe force microscopy (KPFM) as shown in **Figure R6**. Some individual grains are clearly distinguishable, and grain boundaries have higher surface potential than that of the particle interior, which pushes charge carriers away from the grain boundaries and drifts them into particle interior. The topography, contact potential, and their overlapped 3D maps of a $3 \times 3 \mu\text{m}^2$ area (**Figure R7a, b, and c**) and a $1 \times 1 \mu\text{m}^2$ area (**Figure R7d, e, and f**) provide information of the contact potential difference between the grain boundaries and the crystals. We note that our contact potential data are different from those previously reported for the pure perovskites, whose grain boundaries typically have lower contact potential values than that within the grains (*J. Phys. Chem. Lett.* 2015, 6, 875–880). This is because the TFA⁻ anions are abundant at the grain boundaries and can push the charges into

the grains. Therefore, we can conclude that the TFA⁻ ions are accumulated at the grain boundaries.

Figure R6 from **Fig. S10** of our revised manuscript. (a) Topography map and (b) contact potential difference image of the TFA-derived CsPbBr₃ films.

Figure R7 from **Fig. S11** of our revised manuscript. (a,d) Topography, (b,e) contact potential difference, and (c,f), their overlap 3D images for differently sized TFA-derived CsPbBr₃ films.

2. Although the author gives some explanation about the stability of the device enhanced by the TFA, the expression is not so convinced. As we all known, the TFA is a short organic materials. Its own stability is not very good, compared to Cs. So the evidence of the enhancement of the stability should be strengthened in the MS.

Reply:

The TFA is a small organic molecule with a covalent bond, which is relatively stable as compared with many ionic compounds constituting some forms of the perovskites, such as MABr we already mentioned above. In this work, we put forward the point that the ion-migration induced phase-separation results in a severe performance deterioration of PeLEDs. As the TFA ions are able to accumulate at the grain boundaries of perovskite films, this suppresses the ion-migration, and thus enhances the device stability. Please refer to **Figures R8-10**.

Figure R8 from **Fig. 6d** of our revised manuscript. *J-V* hysteresis of PeLEDs based on CsBr(1.7)- and CsTFA(1.7)-derived CsPbBr₃ perovskite films.

Figure R9 from **Fig. S15** of our revised manuscript. Normalized PL spectra of the CsBr- and CsTFA-derived CsPb(Br/I)₃ devices under a constant bias of 2 V. PL spectra were recorded from 0 min to 3 min.

Figure R10 from **Fig. S16** of our revised manuscript. Normalized EL spectra of (a) the CsBr- and (b) CsTFA-derived CsPb(Br/I)₃ devices. EL spectra were recorded from 5V to 8V.

3. *The authenticity of the cross-sectional view is heavily masked by the added color, the contact surface between PEDOT: PSS and perovskite are so smooth. These would misunderstand the readers. The author should revise it.*

Reply:

We agree with this point, and we now have provided original TEM image without using of any artificial colorings. Please see **Figure R12 (Fig.6a in the revised manuscript)**.

Figure R12 from **Fig. 6a** of our revised manuscript. Cross-sectional TEM image of the multi-layer PeLEDs: ITO/PEDOT:PSS (40 nm)/CsPbBr₃ (200 nm)/TPBI (20 nm)/LiF (1 nm)/Al (100 nm).

4. The introduction of FA and MA in perovskite films increased the performance of corresponding devices, while if it can reduce the stability as both MA and FA are extremely moisture-sensitive. Further, please analysis the reason for the development of device performance with FA and MA, especially the beneficial effect of CsTFA.

Reply:

There have been several recent studies performed on the mixed-cation MA/FA perovskite film solar cell devices, which have shown strong improvements of their stability, as compared to MA-only analogues (*1. Energy Environ. Sci. 2016, 9, 1706–1724. 2. Energy Environ. Sci. 2016, 9, 1989–1997.*)

The reasons for the performance enhancement are as follows: i) Doping with larger A-site cations will make the tolerance factor closer to 1, resulting in a more stable device structure and less lattice distortion related trap states (*Scientific Reports*, 2016, 6, 23592); and ii) introducing organic component into inorganic perovskites will change their conductivity from n-type to near-ambipolar, resulting in more balanced charge transport and flatter band conditions under operation.

We have also have shown that our strategy of controlling the perovskite grain growth using TFA is effective not only for inorganic perovskite films but also for organic-inorganic hybrid perovskite films. After TFA treatment, the stability of $\text{FA}_{0.11}\text{MA}_{0.10}\text{Cs}_{0.79}\text{PbBr}_3$ LEDs has been obviously improved as compared with $\text{FA}_{0.11}\text{MA}_{0.10}\text{Cs}_{0.79}\text{PbBr}_3$ LEDs without TFA treatment.

5. Author says that the charge carriers within the CsTFA-derived films are easier to bound together and form excitons, but author do not prove that the excitons are effective for radiative recombination, as excitons may have many behaviors, such as Auger recombination.

Reply:

The flatter energy landscape and better optical properties (**Figure R13**) of CsTFA-derived films show that most of the excitons decay over radiative recombination channel. Several previous studies have shown, that perovskite nanocrystals are less prone to the Auger recombination (*Nano letters*, 2016, 16, 6425), which is not much dependent on the excitation power (*Angewandte Chemie*, 2015,

127, 15644); this makes perovskites rather different from traditional semiconductor materials.

Figure R13 from Fig. 4 of our revised manuscript.

6. Whether such a thick perovskite emitter ~ 200 nm will adversely affect the electrical transport properties of the device.

Reply:

Since the perovskites have higher charge carrier mobilities than organic and covalent semiconductors, the emissive layers of perovskite LEDs can be thicker indeed without compromising the transport properties of these devices. In fact, such

thick perovskite emitter layers have been widely adopted in perovskite LEDs, for example:

1. Lee et al. first reported high-performance perovskite LEDs used a 400 nm thick perovskite emitter (*Science* **350**, 1222–1225 (2015)).

2. Cao et al. reported the highest all-solution-processed perovskite LEDs with record performance used a ~300 nm thick perovskite emitter (*Adv. Mater.* **2018**, *30*, 1804137).

3. Di et al. demonstrated perovskite-polymer bulk heterostructure LEDs exhibiting record-high EQEs exceeding 20% used ~200 nm thick perovskite emitter (*Nat. Photon.* <https://doi.org/10.1038/s41566-018-0283-4> (2018).)

7. *The authors claimed that the PL and EL shifts of mixed-halide CsPb(Br/I)₃ device caused by different time and voltage are attributed to ion migration, and how the authors determined that this phenomenon was not caused by phase separation. Other characterization technique such as TOF need be further probed.*

Reply:

We believe that we are actually talking about the same kind of phenomenon, as the phase segregation is the result of halide migration (*ACS Energy Lett.* **2016**, *1*, 1199–1205). The PL evolution over time under constant voltage, and the EL evolution over applied bias are strong proofs that the ion migration (and the phase separation as a result) is suppressed.

8. *This manuscript needs to be revised in more detail, there are still some errors. For*

example:

i: In the sentence “The strong interaction between DMSO with Pb^{2+} ions and Cs^+ ions is evidenced from the Fourier transform infrared (FTIR) spectroscopy (Fig. 1b)” (page 6, paragraph 2), Fig.1b should be corrected to Fig1c.

ii: In page 21, “Besides, TFA-derived mixed cation $FA_{0.11}MA_{0.10}Cs_{0.79}PbBr_3$ LEDs were fabricated and have shown have shown a maximum luminance of $35,700\text{ cd m}^{-2}$ ”, it have double “have shown”.

Reply:

We appreciate that you brought our attention to these mistakes; all these errors, and the whole manuscript have been carefully revised.

Reply to Reviewer #3

The authors have almost addressed my concerns, but the explanation for improving the device stability is still not very convincing, especially the response for the Q3 mentioned by the Reviewer 3#. The paper could be accepted for publication in present form, but it could be much better if further improvement could be done.

Reply:

We appreciate the overall positive opinion of the reviewer. As for discussion on the improving device stability, we can conclude that the high-quality inorganic perovskite films and the suppressed ion migration may contribute to further improvements of the device stability. We believe that the points we raised in this work, such as more stable crystal structure, flatter energy landscapes and impeded ion motion, will be helpful for many researchers and enable the further progress.

REVIEWERS' COMMENTS:

Reviewer #2 (Remarks to the Author):

This version has been well revised.

This work will have important impact on the field.